# RNP components condense into repressive RNP granules in the aging brain

Kavya Vinayan Pushpalatha [1], Mathilde Solyga [1], Akira Nakamura [2,3] & Florence Besse [1✉]

Cytoplasmic RNP condensates enriched in mRNAs and proteins are found in various cell types and associated with both buffering and regulatory functions. While a clear link has been established between accumulation of aberrant RNP aggregates and progression of aging-related neurodegenerative diseases, the impact of physiological aging on neuronal RNP condensates has never been explored. Through high-resolution imaging, we uncover that RNP components progressively cluster into large yet dynamic granules in the aging *Drosophila* brain. We further show that age-dependent clustering is caused by an increase in the stoichiometry of the conserved helicase Me31B/DDX6, and requires PKA kinase activity. Finally, our functional analysis reveals that mRNA species recruited to RNP condensates upon aging exhibit age-dependent translational repression, indicating that co-clustering of selected mRNAs and translation regulators into repressive condensates may contribute to the specific post-transcriptional changes in gene expression observed in the course of aging.

[1] Université Côte d'Azur, CNRS, Inserm, Institut de Biologie Valrose, Nice, France. [2] Department of Germline Development, Institute of Molecular Embryology and Genetics, Kumamoto University, Kumamoto, Japan. [3] Graduate School of Pharmaceutical Sciences, Kumamoto University, Kumamoto, Japan. ✉email: Florence.BESSE@univ-cotedazur.fr

Formation of membrane-less condensates enriched in functionally related biological molecules has recently emerged as a major principle enabling dynamic cell compartmentalization[1]. As they can selectively and reversibly concentrate molecules, biological condensates are very effective in both buffering intracellular fluctuations and locally regulating molecular activity in response to physiological or environmental changes[1–3]. Ribonucleoprotein (RNP) condensates (or granules), defined by their enrichment in RNA molecules and regulatory proteins, have been observed in the cytoplasm of a variety of species and cell types[4–11], where they are thought to regulate functions ranging from RNA storage and decay to spatiotemporal control of RNA translation and localization[6,10,12–14].

Recently, extensive work has been performed to unravel the molecular principles underlying RNP condensate assembly and dynamic regulation. Studies performed in in vitro reconstituted systems, on one hand, have revealed that purified RNAs and/or proteins condensate into droplets through liquid-liquid phase separation, a process critically dependent on both component concentration and the establishment of dense networks of protein-protein and protein-RNA interactions[15–18]. Studies performed in living cells, on the other hand, have shown that cytoplasmic RNP condensates exhibit under normal conditions properties expected from liquid-like entities, including high component turnover as well as inter-condensate fusion and mixing[3,19–24].

As further revealed by proteomic and transcriptomic analyses, endogenous RNP condensates have a complex composition characterized by the presence of up to hundreds of proteins and RNA molecules[25–30]. These molecules are not all functionally equivalent: while a limited number of resident molecules, referred to as scaffolds, are required for the nucleation of RNP condensates, others, termed clients, are dispensable and recruited in a context-dependent manner[31,32], raising the question of how condensate molecular specificity is encoded. Recent ex vivo work revealed that both post-translational modification of RNP components[33–36] and competition between RNA and protein interaction networks[34,37,38] are key drivers of differential sorting. If and how these principles are used by biological systems to modulate RNP granule properties and function in response to physiological cues still remains unclear.

In the healthy nervous system, collections of dynamic RNP granules with both distinct and overlapping composition have been identified[9,10,27,39,40]. In neurodegenerative diseases such as Amyotrophic lateral sclerosis (ALS) or Frontotemporal Degeneration (FTD), in contrast, accumulation of static RNP aggregates with aberrant composition has been reported, suggesting that such pathological aggregates might trap RNA binding proteins (RBPs) and be involved in disease progression[41–43]. As these neurodegenerative diseases typically have a late onset, these findings led to the suggestion that aging may be an important factor contributing to RNP aggregation[44–46]. To date, however, if and how physiological aging impacts on neuronal RNP condensates independently of disease context has remained unexplored.

To address this question, we analyzed in vivo how RNP granule properties are regulated in the context of the aging *Drosophila* brain. We uncovered that cytoplasmic RNP components progressively condense into large yet dynamic granules upon physiological aging, a process accompanied by a decrease in the diversity of RNP granule composition, but distinct from the age-dependent aggregation observed in neurodegenerative contexts. Combining quantitative imaging and genetics, we further discovered that aging neurons regulate RNP component condensation both by increasing the stoichiometry of the conserved DEAD-box helicase Me31B/DDX6/Rck and through PKA-dependent phosphorylation. Functionally, age-dependent

remodeling of RNP granules associates with translational repression of mRNAs recruited to granules, indicative of a repressive function for neuronal RNP granules. Together, this work illustrates how the key parameters underlying condensate assembly are tuned in vivo by biological systems and shows how this connects to the regulation of RNA fate. This study also functionally demonstrates that aging, independently of associated diseases, impacts on the in vivo properties and function of RNP condensates, opening new perspectives on the regulation of gene expression in the context of aging brains.

## Results

**Increased condensation of neuronal RNP components upon aging.** To monitor the impact of physiological aging on neuronal RNP granule properties, we first analyzed the subcellular distribution of known granule markers in *Drosophila* brains of two different ages that we subsequently refer to as young and aged, respectively: 1–2-days post-eclosion (i.e. right after the extensive neuronal maturation occurring upon eclosion) and 35–39 days post-eclosion (i.e. at about mid-life, before significant drop in fly viability). The DEAD-box helicase Me31B/DDX6/Rck and the RNA binding protein (RBP) Imp/ZBP-1/IGF2BP were chosen as markers, as both are conserved RNA-associated proteins known to localize to RNP granules in vertebrate and invertebrate neurons[47–52]. In *Drosophila*, cytoplasmic Me31B-positive and Imp-positive RNP granules have previously been described in the soma of Mushroom Body γ neurons, a population of neurons essential for learning and memory functions[24,47,48,53–55]. In young brains, indeed, we observed that endogenous Me31B and Imp accumulate into numerous small punctate structures (Fig. 1a, c), a distribution recapitulated in Me31B-GFP and GFP-Imp knock-in lines (Fig. 1e, g). These two proteins however displayed very different partitioning properties at this stage. Me31B was mostly found in granules (Fig. 1a, e), exhibiting a high partition coefficient calculated as the intensity ratio between the granule-associated and the soluble cytoplasmic pools (Fig. 1k). In contrast, a significant fraction of Imp localized diffusively throughout the cytoplasm (Fig. 1c, g), resulting in a lower partition coefficient (Fig. 1k). Remarkably, increased condensation of Me31B and Imp into large granules was observed in aged brains (Fig. 1b, d, f, h), a process characterized by a significant increase in both granule size (Fig. 1l; Supplementary Fig. 1a) and intra-granule concentration of Me31B and Imp (Fig. 1n). Increased condensation was particularly visible for Imp whose diffuse cytoplasmic signal strongly decreased (Fig. 1d, h), resulting in an increase in the number of Imp-enriched granules detectable over the cytoplasmic diffuse pool (Supplementary Fig. 1b and Fig. 5e). In contrast, the number of Me31B-enriched granules decreased, as Me31B tended to condense into fewer, but larger granules (Fig. 1l, m). To determine if the observed condensation of RNP component reflected an abrupt, or rather a more gradual change in RNP component distribution, we next analyzed brains at three additional time points after eclosion: 10 days, 20–23 days and 50–52 days. As shown in Fig. 1o and Supplementary Fig. 1b, clustering was already visible in 10 day-old brains and continuously increased from 10 days to 50 days post-eclosion, arguing against a sudden switch in behavior.

To then test if the re-distribution of Me31B and Imp observed upon aging reflects a general trend, we analyzed the localization of other conserved neuronal granule components including Trailer-hitch/Lsm-14 (Tral), HPat1 and Staufen (Stau)[39,56,57]. As observed for Me31B and Imp, these proteins condensed into larger cytoplasmic granules upon aging (Supplementary Fig. 1c–h, k, l). Clustering, however, was not observed for stress granule components such as Rin/G3BP (Fig. 1i, j) or PABP (Supplementary Fig. 1i, j),

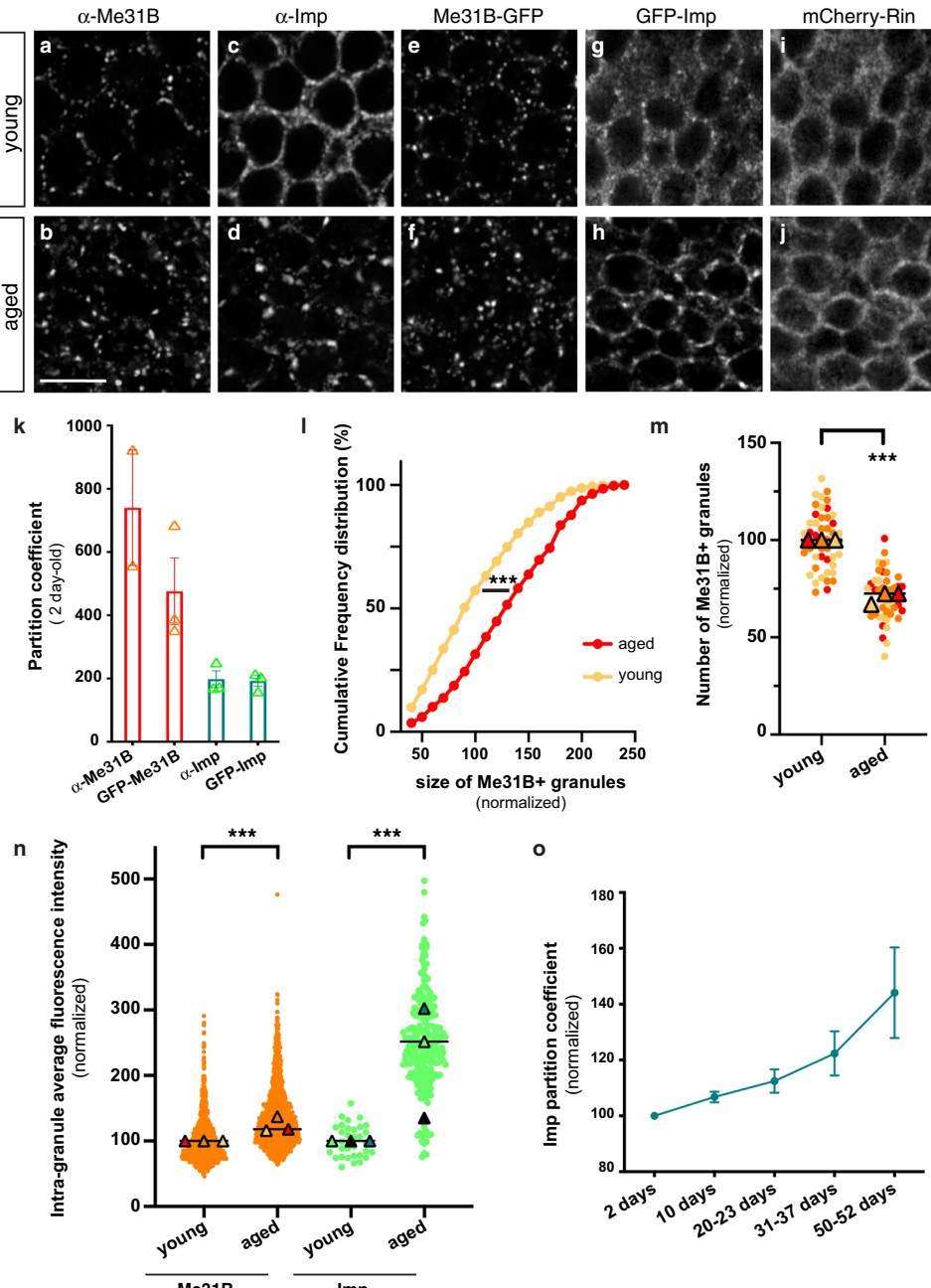

**Fig. 1 Me31B and Imp condensate into larger cytoplasmic granules in aged brains. a–j** Cell bodies of MB γ neurons imaged from 1–2 day- (**a**, **c**, **e**, **g**, **i**; young), or 37–38 day- (**b**, **d**, **f**, **h**, **j**; aged) old brains. Brains were stained with anti-Me31B (**a**, **b**) or anti-Imp (**c**, **d**) antibodies. Me31B-GFP (**e**, **f**), GFP-Imp (**g**, **h**) and mCherry-Rin (**i**, **j**) were expressed from their endogenous locus. Note that dark nuclei occupy most of the soma, and that the cytoplasmic signal is restricted to the cell periphery. Scale bar: 5 μm. **k** Mean granular:cytoplasmic intensity ratio (partition coefficient) in brains of 1–2 day-old flies. Each data point represents the mean value of one replicate. At least 10 fields were analyzed per replicate. Two to three independent experiments were quantified per condition. **l** Normalized sizes of Me31B-containing granules in neurons of 1–2 day- (young) and 37–38 day- (aged) old brains. Frequency distributions are plotted for one replicate, but three were performed (see Supplementary Fig. 1a). ***$P < 0.001$ (unpaired t-test on individual data points). **m** Normalized numbers of Me31B-containing granules (per surface area) in MB γ neurons of 1–2 day- (young) and 37–38 day- (aged) old brains. Three replicates were performed and the mean value of each is indicated as a triangle. Data points were color-coded based on the replicate they belong to. In **l**, **m**, individual data points were collected from brains immunostained with anti-Me31B antibodies and normalized to the young condition. ***$P < 0.001$ (unpaired, two-sided $t$-test on individual data points). **n** Intra-granule concentration of Me31B (orange) and Imp (green) in 1–2 day- (young) and 37–38 day- (aged) old brains. Concentrations were estimated by measuring the mean intensity of Me31B-GFP and GFP-Imp signals within masks of corresponding granules. Three replicates were performed and the mean value of each is indicated as a triangle. The distribution of individual granule mean intensities is shown for one replicate only. ***$P < 0.001$ (unpaired, two sided $t$-test on individual data points). **o** Imp mean partition coefficients upon aging. Each data point represents the average of mean values obtained from three independent replicates. Error bars correspond to s.e.m. Source data are provided as a Source Data file.

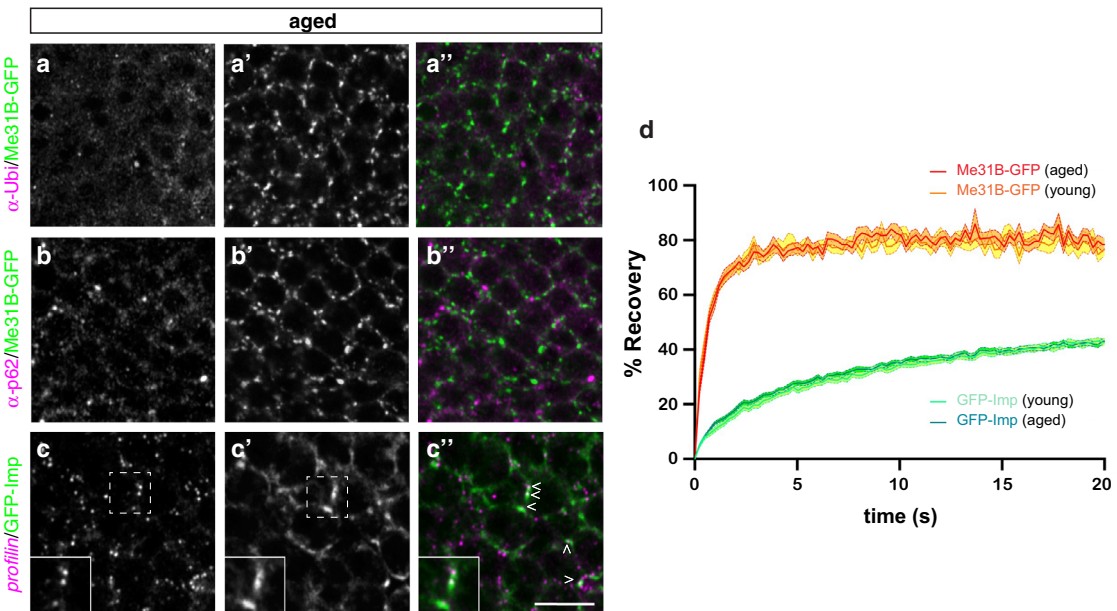

**Fig. 2 Large granules in old flies are dynamic RNP assemblies. a–c** Cell bodies of MB γ neurons imaged from 37–38 day-old (aged) brains. Me31B-GFP-expressing brains were stained with anti-Ubiquitin (**a**, magenta in **a''**) antibodies, anti-p62 (**b**, magenta in **b''**) antibodies, or *profilin* smFISH probes (**c**, magenta in **c''**). Me31B-GFP distribution is shown in white in **a'**-**c'** and in green in **a''**-**c''**. Arrowheads in **c''** point to large granules containing *profilin* mRNA. Insets show magnifications of the boxed region in **c,c'**. Two independent biological replicates were performed. Scale bar: 5 μm. **d** Average FRAP curves obtained after photobleaching of GFP-Imp-positive (green) or Me31B-GFP-positive (red-orange) granules from 1–2 day- (young) or 38 day- (aged) old brain explants. A minimum of 90 granules from at least 9 brains were analyzed per condition. Error bars (fill areas) represent s.e.m. Source data are provided as a Source Data file.

indicating first that not all RBPs tend to cluster upon aging, and second that large granules do not correspond to entities forming in response to stress.

**The large neuronal RNP granules observed in aged brains are dynamic assemblies**. Protein aggregates have been observed in brains of aged animals and human patients in response to altered proteostasis[58]. To determine if the large granules observed in aged brains correspond to protein aggregates, we first performed co-localization experiments using antibodies recognizing p62/Ref(2) P and Ubiquitin, both known to accumulate in protein aggregates found in aged flies[59]. As shown in Fig. 2a-a'', b-b'' and Supplementary Fig. 2a-a'', b-b'', p62 + and Ubiquitin + aggregates were visible in aged brains, but did not co-localize with the large Imp- and Me31B-positive granules found at this age. To further exclude that Me31B and Imp get enriched into insoluble aggregates upon aging, we collected the soluble and insoluble fractions of young and aged brains through cell fractionation of head lysates. No significant increase in the amount of Me31B or Imp sedimenting in the insoluble fraction was observed upon aging (Supplementary Fig. 2c). Last, to test if the large granules contained RNA, we performed smFISH experiments with probes recognizing *profilin (prof)* mRNA, that we previously identified as a direct target of Imp[60]. As shown in Fig. 2c-c'' (arrowheads), *profilin* transcripts could be found in large Imp-positive granules in aged brains, further indicating that these entities correspond to bona fide neuronal RNP granules.

In in vitro reconstituted systems, RNP droplets mature over time into less dynamic entities, transitioning from liquid to more solid-like states[21,45,61]. To test if in vivo physiological aging is associated with changes in the turnover of RNP granule components, we performed FRAP experiments on intact young and aged brains and compared fluorescent signal recovery upon bleaching of Me31B-GFP + or GFP-Imp + granules. Remarkably, the two RNP components exhibited very different recovery

rates, with Me31B-GFP signal exhibiting a near complete recovery within few seconds and GFP-Imp signal exhibiting only partial (~40%) recovery after dozens of seconds (Fig. 2d)[48]. Similar recovery, however, was observed for each protein in young and aged brains, indicating that the dynamic turnover of granule-associated proteins is not affected by aging.

**Me31B and Imp condense into common multiphase condensates upon aging**. Previous biochemical and imaging studies have shown that collections of RNP granules with partially overlapping, but distinct composition are typically observed in neuronal cells[9,10,27]. Consistent with this, we found that Imp is recruited to significant levels only in a subset of Me31B + granules in young MB γ neurons (Fig. 3a-a''), resulting in the co-existence of both Me31B + Imp + (white arrowheads in Fig. 3a-a'') and Me31B + Imp- (blue arrowheads in Fig. 3a-a'') granules. To determine if the differential recruitment of Imp to Me31B + granules was modified upon aging, we compared the relative proportion of Me31B + Imp + and Me31B + Imp- granules in young and aged brains. As shown in Fig. 3b-b'', c, a more than 2-fold increase in the proportion of Me31B + Imp + granules was observed upon aging, with almost all Me31B + granules containing Imp in aged brains. As further revealed by high-resolution imaging of fixed (Fig. 3d) or living (Supplementary Movie 1) brains, Imp and Me31B did not homogenously mix in Imp + Me31B + granules, but rather segregated into distinct sub-domains within neuronal RNP condensates. Together, these results thus indicate that Me31B and Imp systematically condense into common multiphase RNP granules upon aging, thus reflecting age-dependent changes in the differential recruitment of RNP components to distinct granules.

**Age-dependent increase in Me31B levels induces its condensation**. As the composition of condensates is known to

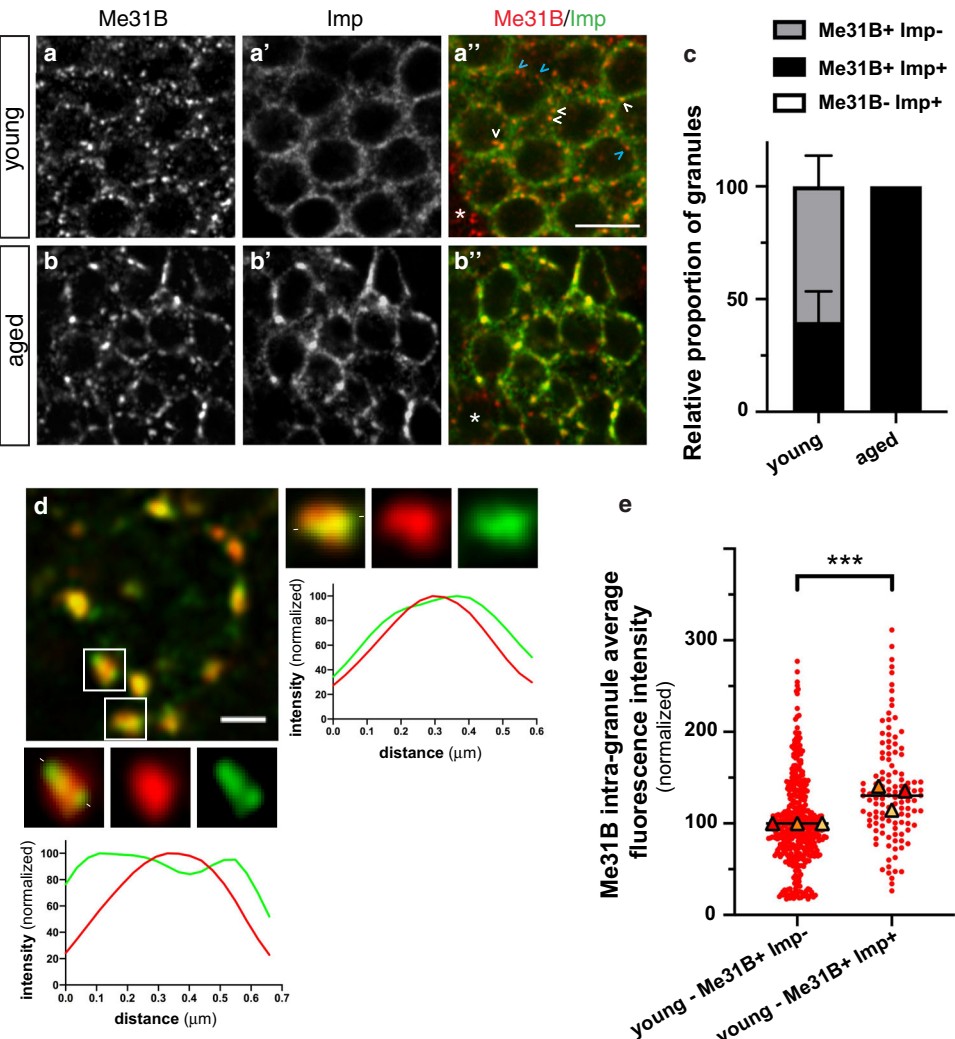

**Fig. 3 Age-dependent changes in RNP granule composition. a, b** Cell bodies of MB γ neurons imaged from 1–2 day- (**a**, young) or 37–38 day- (**b**, aged) old brains. GFP-Imp- (gray in **a'**,**b'** and green in the overlay) expressing brains were stained with anti-Me31B antibodies (gray in **a**, **b** and red in the overlay). The white arrowheads point to Me31B + Imp + granules while the blue ones point to Me31B + Imp- granules. Asterisks in **a"**, **b"** indicate non-MB γ neurons that do not express Imp. Scale bar: 5 μm. **c** Relative proportions of Me31B + Imp- and Me31B + Imp + granules in young and aged flies. The fraction of Me31B + granules containing Imp was estimated using the JACoP plugin of Fiji (see Method), and values normalized to the aged condition. Three replicate experiments, with at least 10 samples per condition were performed and averaged. Error bar represents s.e.m. Note that no Me31B − Imp+ granules could be observed. **d** Left: cell body of a 38 day-old MB γ neuron expressing GFP-Imp (green) and stained with anti-Me31B antibodies (red). Scale bar: 1 μm. Magnifications of the boxed areas are shown in the lower and right panels. Intensity profiles of GFP-Imp (green) and Me31B (red) measured along the line marked by white segments are shown below magnified images. **e** Concentration of Me31B-GFP in granules negative (left) or positive (right) for Imp in 1–2-day-old brains. Three replicates were performed and the mean value of each replicate is indicated as a symbol (triangle). Data points were color-coded based on the experimental replicate they belong to and normalized to the values found in Me31B + Imp- granules. ***$P < 0.001$ (unpaired, two-sided $t$-test on individual data points). Source data are provided as a Source Data file.

depend on the relative stoichiometry of their constituent molecules[31,32], we then investigated whether the differential recruitment of Imp to Me31B + granules may correlate with the concentration of Me31B molecules in the distinct granules found in young brains. Plotting Me31B signal intensity in Imp + and Imp- granules revealed that Me31B + Imp + granules exhibited a significantly higher concentration of Me31B than Me31B + Imp- granules (Fig. 3e), suggesting that Me31B might play an important role in the recruitment of Imp. To further test this hypothesis, we inactivated *me31B* through RNAi, which resulted in the near absence of Me31B protein (Supplementary Fig. 3a–c). This inhibited the recruitment of Imp to granules (Supplementary Fig. 3b–d) without affecting overall Imp protein levels (Supplementary Fig. 3e). These results, together with the increased

condensation of Me31B observed upon aging, suggested that the intra-granule concentration of Me31B might be an important determinant underlying age-dependent RNP remodeling.

Changes in the intra-granule concentration of RNP components can be induced by changes in partitioning properties and/or by changes in overall component concentration[62]. To investigate the origin of the age-dependent changes in Me31B behavior, we thus measured both Me31B partition coefficient and Me31B global amount in young and aged MB neurons. Remarkably, while no increase in Me31B partition coefficient was observed upon aging (Fig. 4a), a 50% increase in Me31B protein levels was observed (Fig. 4b). Such a dosage increase was not observed for Imp (Fig. 4b). Furthermore, it did not correlate with increased *me31B* RNA levels (Fig. 4c), suggesting the existence of post-transcriptional regulatory

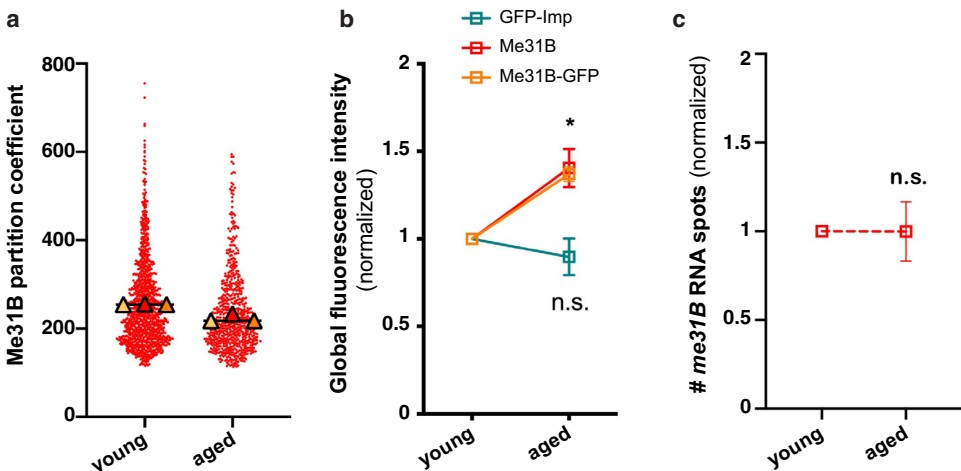

**Fig. 4 Me31B levels increase upon aging, a process regulated at the post-transcriptional level. a** Partition coefficient of Me31B-GFP in 1–2 day- (young) and 37–38 day- (aged) old brains. Partition coefficients were estimated by dividing the maximal intensity of Me31B signal in individual RNP granules to the intensity of the cytoplasmic diffuse pool (see Methods), and calculated for each granule detected in the imaged fields. Three replicates were performed and the mean value of each replicate is indicated as a symbol (triangle). The distribution of individual granule partition coefficients is shown for one replicate only. **b** Global Imp and Me31B levels measured from confocal images of 1–2 day- (young) and 37–40 day- (aged) old MB γ neurons. Imp levels were measured using endogenously expressed GFP-Imp fusions (green). Me31B levels were measured using both endogenously-expressed Me31B-GFP fusions (orange) and α-Me31B antibodies (red). *$P < 0.05$ (Kruskall–Wallis test; $n = 3$; $P = 0.02$). **c** Normalized numbers of *gfp* smFISH spots counted in 1–2 day- (young) and 37–38 day- (aged) old Me31B-GFP-expressing brains. n.s. stands for not significant (unpaired, two-sided Mann–Whitney test on replicate means). In **b**, **c**, data points represent the mean values obtained from three independent replicates. Error bars correspond to s.e.m. Values were normalized to the young condition. At least 10 fields were imaged per replicate and per condition. *$P < 0.05$. Source data are provided as a Source Data file.

mechanisms that modulate the levels of Me31B in the course of aging and may cause increased RNP condensation.

To then functionally test if reducing the dosage of Me31B would prevent age-dependent condensation of neuronal RNP components, we genetically removed a copy of me31B using two previously described deletions (*me31BΔ1* and *me31BΔ2*)[63]. Reducing by half the total amount of Me31B (Supplementary Fig. 4a, b) induced a significant decrease in Me31B intra- granule concentration in aged brains (Fig. 5a) without changing Me31B partition coefficient (Supplementary Fig. 4c). Furthermore, it limited both the size (Fig. 5b–d and Supplementary Fig. 4d) and the number (Supplementary Fig. 4e) of Me31B-positive granules, such that values found in aged *me31BΔ1/ +* and *me31BΔ2/ +* brains were similar to those of young control brains. In contrast to what was observed when abolishing *me31B* expression (Supplementary Fig. 3a), partially reducing the dosage of *me31B* only mildly decreased the age-dependent condensation of Imp, as illustrated by the still elevated number of Imp-enriched granules detected in aged *me31BΔ1/ +* and *me31BΔ2/ +* brains (Fig. 5c', e and Supplementary Fig. 4f). Together, these results demonstrate that the concentration of Me31B increases upon physiological aging, triggering its condensation into large granules. Increasing the stoichiometry of Me31B only mildly promoted the recruitment of Imp to Me31B-containing granules, though, indicating the existence of complementary molecular pathways regulating Imp condensation.

**PKA is required for the condensation of Imp in aged flies.** Having shown that Me31B dosage only partially impacts on Imp condensation, we sought to identify pathways that may regulate this process. To this end, we performed a selective screen in which the activity of conserved pathways known to either be impacted by aging, or contribute to aging was modulated[58,64]. RNAi or dominant negative constructs were expressed specifically in adult MB neurons to avoid developmental contributions, and the subcellular distribution of Imp analyzed in aged flies. Strikingly, altering most of the pathways tested did not have a significant impact on Imp

condensation (Supplementary Table 1). Inhibiting the activity of the cAMP-dependent kinase PKA via expression of a catalytic-dead variant, however, prevented the condensation of Imp into granules (Fig. 6a, b, d) without affecting Imp levels (Supplementary Fig. 5c). Similar results were observed when expressing dsRNA targeting PKA catalytic domain (PKA-C1), or when inactivating *amnesiac*, a gene known to produce a peptide activating PKA in MB neurons (Fig. 6c, d)[65]. Remarkably, PKA inhibition did not significantly affect Me31B levels (Supplementary Fig. 5c) and only mildly impacted the condensation of Me31B (Fig. 6e and Supplementary Fig. 5a, b). Furthermore, it did not alter the condensation of other RNP components including Tral and HPat1 (Supplementary Fig. 6a, b), suggesting that PKA does not generally modulate the behavior of RNP components, but rather specifically targets Imp. As in vivo inactivation of PKA may have long-term indirect effects, we used a complementary ex vivo system, in which brain explants of aged flies were acutely treated with the H-89 PKA inhibitor. As shown in Supplementary Fig. 6c, d, treatment with H-89 induced within dozens of minutes the decondensation of Imp and its relocalization from the granular to the diffuse cytoplasmic pool. Together, these results thus demonstrate that the activity of PKA is dispensable for Me31B condensation, but is required acutely and selectively for the recruitment of Imp to RNP granules in aged brains.

**The granule-associated *profilin* RNA species gets translationally repressed upon aging.** Neuronal RNP granules are enriched in translational repressors and thought to contain translationally-repressed mRNAs[9,27,66,67]. As not only Imp, but also its target RNA *profilin (prof)*, showed increased (although not complete) recruitment to Me31B-positive granules upon aging (Fig. 7a and Supplementary Fig. 7a–c), we wondered if increased RNP component condensation was accompanied by the translational inhibition of mRNA species found in granules. To this end, we expressed an inducible translational reporter generated by fusing the coding sequence of EGFP to the 3'UTR of *profilin* and used SV40 3'UTR as

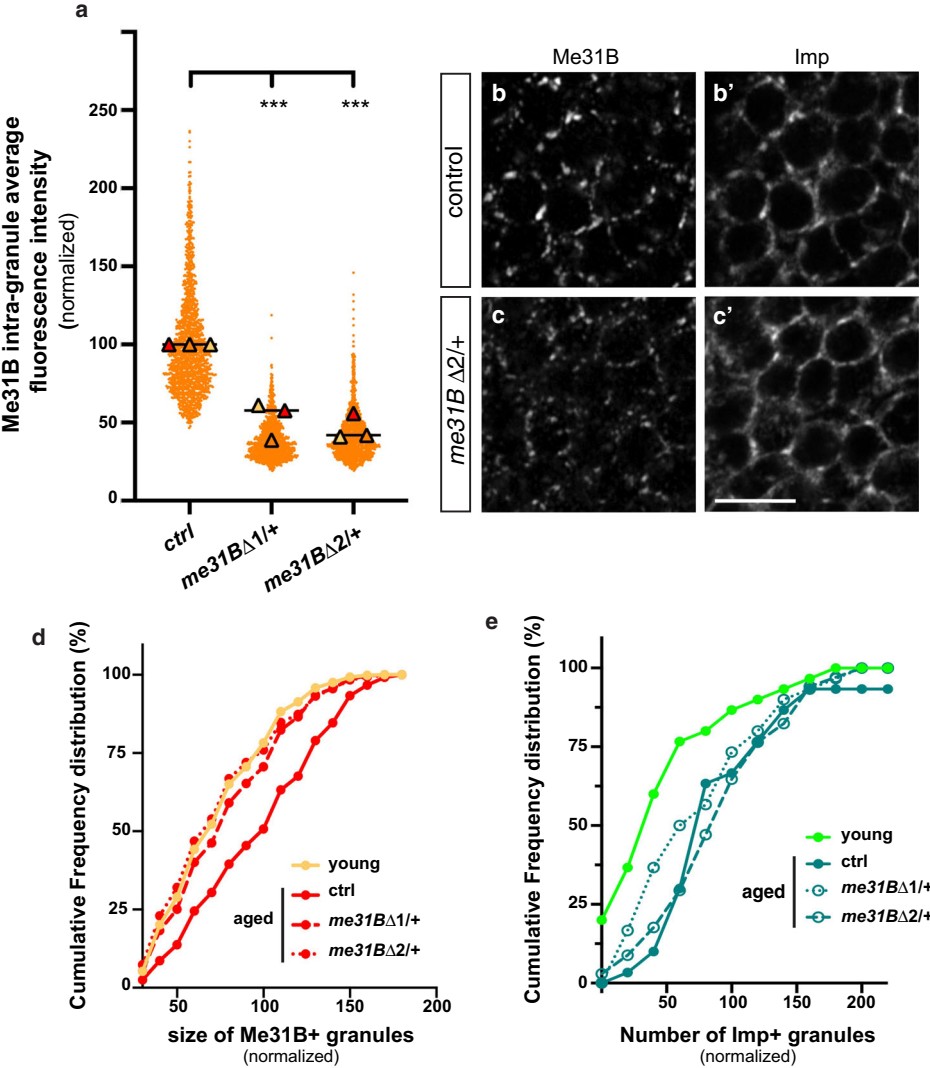

**Fig. 5 Reducing Me31B levels decreases Me31B condensation. a** Intra-granule concentration of Me31B-GFP in granules of 37–38 day-old *me31B Δ1/* + and *me31BΔ2/* + MB γ neurons. Three replicates were performed and the mean value of each replicate is indicated as a symbol (triangle). Values were normalized to the control condition. The distribution of individual granule values is shown for one replicate only. ****P* < 0.001 (one-way ANOVA with Dunn's multiple comparison tests). **b, c** Cell bodies of control (**b, b'**) and *me31BΔ2/* + (**c, c'**) MB γ neurons from 37–38 day-old brains stained with anti-Me31B (**b, c**) and anti-Imp (**b',c'**). Scale bar: 5 μm. **d** Sizes of Me31B-containing granules in MB γ neurons of 1–2 day- (light orange; young) and 37–38 day- (red; aged) old brains. Values were normalized to aged controls. Cumulative frequency distributions are plotted for one replicate, but three replicates were performed in total (see Supplementary Fig. 4d). ****P* < 0.001 (one-way ANOVA test). **e** Numbers of Imp + granules (per surface area) in MB γ neurons of 1–2 day- (light green; young) and 37–38 day- (green; aged) old brains. Values were normalized to aged controls. Cumulative frequency distributions are plotted for one replicate, but three replicates were performed in total (see Supplementary Fig. 4f). ****P* < 0.001 (one-way ANOVA test). Source data are provided as a Source Data file.

a negative control. As expected, *gfp-profilin* 3'UTR RNA molecules partially, yet significantly, associated with Me31B + granules and were recruited to RNP granules at higher level in aged compared to young brains (Fig. 7b). Remarkably, quantification of GFP protein levels in young vs aged brains revealed a significant decrease in the amount of GFP expressed from the *profilin* reporter, but not from the *SV40* control, upon aging (Fig. 7e–g). As measured by quantification of *gfp* smFISH signals (Fig. 7c, d and Supplementary Fig. 7d), decreased GFP expression did not correlate with decreased *gfp* RNA levels, indicating that the translation of *profilin* reporter is down-regulated upon aging. To further confirm the specificity of *profilin* reporter behavior, we analyzed the expression of a reporter construct for *camk2*, an RNA known to undergo 3'UTR-dependent regulation[47,68,69], but not enriched in Imp-positive granules (Supplementary Fig. 7a, b). As shown in Fig. 7g, the translation of

GFP-*camk2* 3'UTR did not decrease upon aging, suggesting that translation down-regulation is not a general trend, but rather appears to be specific to RNA species associating with RNP granules.

Last, to confirm that the behavior of *gfp* reporters reflected physiological regulation, we analyzed the expression of endogenous *profilin*. Remarkably, a significant decrease in Profilin protein levels (Supplementary Fig. 8a), but not RNA levels (Supplementary Fig. 8b), was observed upon aging, consistent with translational downregulation. As *profilin* encodes a G-actin binding protein shown to promote the polymerization of F-actin in *Drosophila* tissues[70], we wondered if the observed decrease in Profilin levels might be accompanied by modifications of the actin cytoskeleton and expressed a fluorescently tagged version of the actin-binding and bundling protein Fascin[71]. A significant decrease in somatic

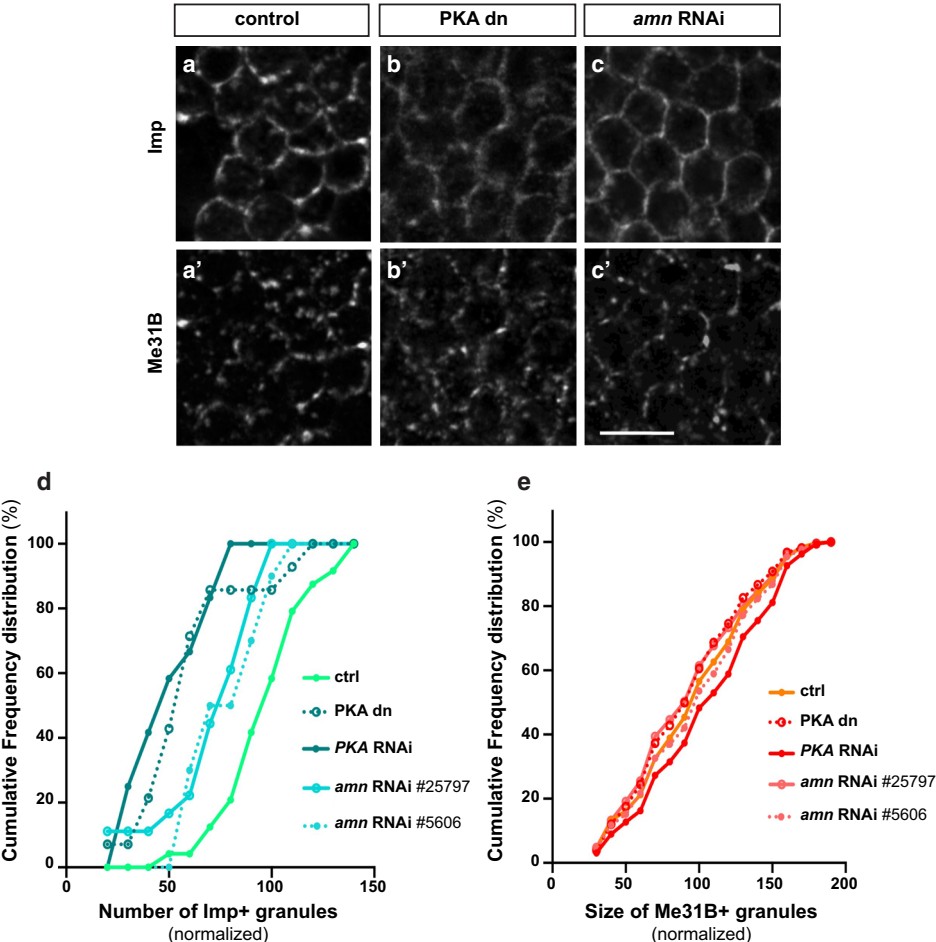

**Fig. 6 Inactivating PKA inhibits the recruitment of Imp to RNP granules in aged flies. a–c** Cell bodies of MB γ neurons imaged from 30 day-old (aged) control brains (**a**), brains expressing a kinase dead PKA catalytic subunit (**b**), or brains expressing *amn* RNAi (**c**). PKA dn stands for PKA dominant negative. Scale bar: 5 μm. **d** Numbers of Imp + granules (per surface area) in MB γ neurons of 30 day-old control brains (light green) or brains with reduced PKA activity (dark green). Values were normalized to aged controls. Cumulative frequency distributions are plotted for one replicate, but three were performed in total (see Supplementary Fig. 5d). **e** Sizes of Me31B-containing granules in MB γ neurons of 30 day-old control brains (light orange) or brains with reduced PKA activity (red). Values were normalized to aged controls. Cumulative frequency distributions are plotted for one replicate, but three were performed in total (see Supplementary Fig. 5a). Source data are provided as a Source Data file.

cortical Fascin signal was observed upon aging (Supplementary Fig. 8c–e), indicating age-related modifications of the F-actin network. Together, these results thus indicate that *profilin* RNA exhibits both increased recruitment to granules and specific translational repression upon aging, a process accompanied by changes in the F-actin cytoskeleton network.

**Recruitment of Imp to Me31B + granules underlies age-dependent translational repression**. To further study the link between RNP component condensation and translational repression, we aimed at assessing GFP-*profilin*−3'UTR expression upon aging in contexts where condensation is altered. To this end, we first inactivated *me31B* via RNAi, which did not impact on GFP-*profilin*−3'UTR expression in young flies (Supplementary Fig. 9a) but significantly inhibited the condensation of Imp in aged brains (Supplementary Fig. 3b–d). Remarkably, GFP-*profilin*−3'UTR expression did not exhibit age-dependent decrease in this context (Fig. 8a), suggesting that condensation may be essential for the age-dependent translational repression of *profilin*.

Decreased repression of *profilin* translation was also observed upon PKA inhibition (Fig. 8b and Supplementary Fig. 9b), a context in which Me31B condensation is preserved, but Imp recruitment to granules altered (Fig. 6d). Such an effect did not

reflect a general function of PKA in translational repression, as PKA inactivation did not increase the expression of control *gfp-SV40*−3'UTR RNAs or non granule-associated RNAs such as *gfp-camk2*−3'UTR (Fig. 8c and Supplementary Fig. 9b). Together, these results thus indicate that the specific recruitment of Imp and target RNAs to RNP granules might promote selective translational repression upon aging.

## Discussion

**Me31B stoichiometry controls neuronal RNP granule properties in vivo**. Our functional inactivation of *me31B* through RNAi has revealed that the conserved DEAD-box helicase Me31B is required for the assembly of neuronal RNP granules recruiting Imp and may thus be considered as a core or scaffold component. Such a function is reminiscent of the role reported for Me31B orthologs DDX6 and Dhh1 in assembling P-bodies, and likely relates to the capacity of Me31B/DDX6/Dhh1 proteins to oligomerize and bind RNA with high affinity yet poor specificity[72,73]. Remarkably, we also uncovered that the concentration of Me31B in RNP granules increases upon aging, a phenomenon that does not result from an increase in the partition coefficient of Me31B, but rather from an increase in the cellular levels of Me31B. How do neurons physiologically modulate the concentration of the RNP scaffold Me31B

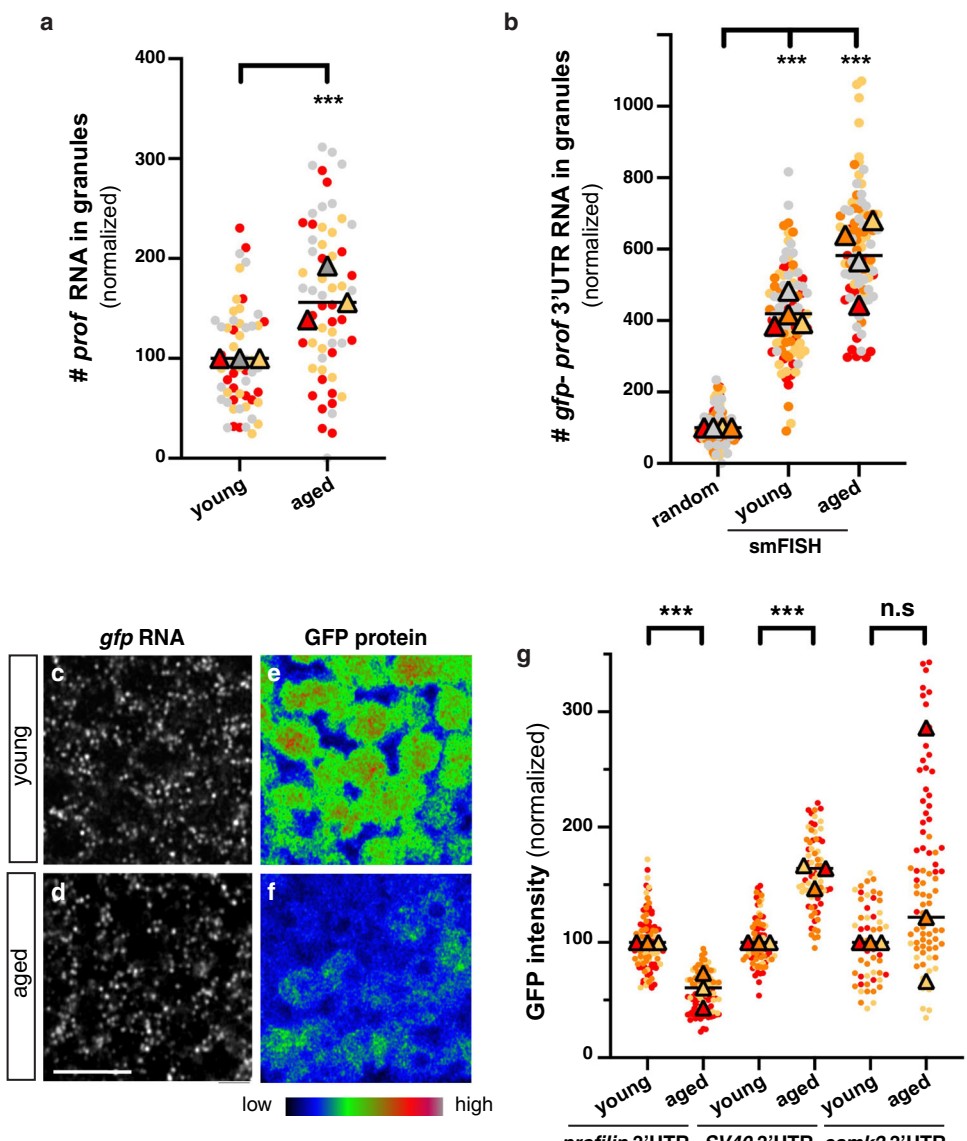

**Fig. 7 Age-dependent decrease in the translation of granule-associated mRNA species. a** Fraction of endogenous *profilin* RNA molecules found in Me31B-GFP + granules of 1–2 day- (young) and 37–38 day- (aged) old flies. Values were normalized to the young condition. ***$P < 0.001$ (unpaired, two-sided *t*-test). **b** Fraction of *gfp-profilin 3'UTR* RNA molecules found in Me31B-mTomato + granules of 1–2 day- (young) and 37–38 day- (aged) old flies. Values were normalized to the random control, in which non-specific levels of co-localization were assessed after rotating smFISH images by 90°. ***$P < 0.001$ (one-way ANOVA with Dunn's multiple comparison tests). In **a**, **b**, the fraction of smFISH spots co-localizing with Me31B + granules was estimated for each imaged field using the JACoP plugin of Fiji (see Method). Three to four replicates were performed and the mean value of each is indicated as a triangle. Data points were color-coded based on the replicate they belong to. At least 16 fields were imaged per condition. **c–f** Cell bodies of MB γ neurons expressing *gfp-profilin 3'UTR* transcripts in 1–2 day- (**c**, **e**; young) or 37–38 day- (**d**, **f**; aged) old brains. smFISH signals obtained using *gfp* probes are shown in **c**, **d**. GFP signals are shown in **e**, **f**. Images in **e**, **f** were color-coded using the Rainbow RGB visualization mode of Fiji. Scale bar: 5 μm. **g** GFP signal intensities measured from 1–2 day- (young) and 37–40 day- (aged) old brains expressing GFP-*profilin* 3'UTR (left), GFP-*SV40* 3'UTR (middle) or GFP-*camk2* 3'UTR (right). Constructs were expressed using the OK107-Gal4 driver. Values were normalized to the young conditions. Three replicates were performed and the mean value of each is indicated as a triangle. Data points were color-coded based on the experimental replicate they belong to. At least 12 fields were imaged per condition. ***$P < 0.001$ (one-way ANOVA test with Sidak's multiple comparison tests, performed on individual data points). n.s. stands for not significant. For the *camk2* 3'UTR reporter, eight outlier data were omitted from the graph (although they were considered to calculate the mean of the corresponding replicate and to perform statistical tests). Source data are provided as a Source Data file.

remains to be addressed, but we have shown that this regulation is post-transcriptional and thus likely reflects increased translation of *me31b* RNA molecules. Modulating the dosage of Me31B has consequences on RNP granule properties, as age-dependent increase in Me31B levels associates with the condensation of Me31B into less but larger granules, and with the recruitment of Imp to all (and not just a subset) of Me31B + granules. Furthermore, partially reducing *me31B* dosage largely inhibits

age-dependent condensation of Me31B, resulting in the maintenance of both Me31B + Imp + and Me31B + Imp- granules in old brains. These results suggest a dose-response model in which client RNP components (*e.g.* Imp) competing for interaction sites are recruited to subsets of granules only at limiting concentrations of Me31B (*i.e.* in young brains), but condense into all Me31B + granules at higher concentrations of Me31B, in conditions favoring the binding of multiple clients to common scaffolds. These results

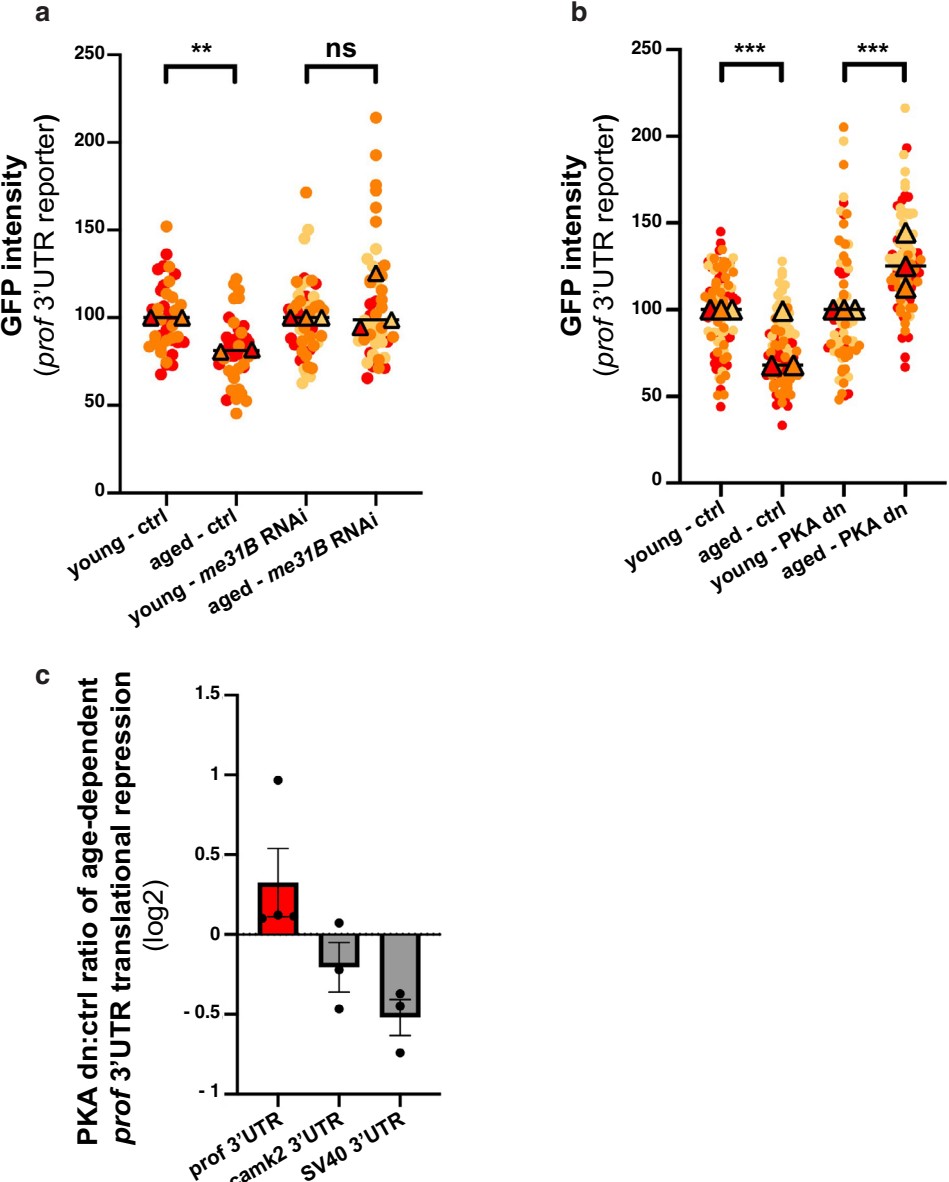

**Fig. 8 Me31B and PKA are required for age-dependent translation repression. a** GFP signal intensities measured from 1–2 day- (young) and 37–38 day-(aged) old brains expressing GFP-*profilin* 3'UTR with (right) or without (left) *me31B* RNAi. Both the GFP-*profilin* 3'UTR reporter and the *me31B* RNAi construct were expressed in MB neurons using the OK107-Gal4. Two replicates were performed for the control condition and three for the RNAi condition. **b** GFP signal intensities measured from brains expressing GFP-*profilin* 3'UTR in control and PKA dn (dominant negative) conditions. Both the GFP-*profilin* 3'UTR reporter and the PKA dn construct were expressed specifically in adult MB neurons using tub-Gal80ts; OK107-Gal4. To prevent Gal4 titration effect, a neutral luciferase construct was expressed together with the *profilin* 3'UTR reporter for the control condition. In **a**, **b**, values were normalized to respective young control flies. Two to three replicates were performed and the mean value of each replicate is indicated as a symbol (triangle). Data points were color-coded based on the experimental replicate they belong to. At least 12 fields were imaged per condition. \*\*P < 0.01; \*\*\*P < 0.001 (one-way ANOVA test with Sidak's multiple comparison tests, performed on individual data points). n.s. stands for not significant. In **a**, P = 0.0012 for controls and P = 0.09 for *me31B* RNAi condition. In **b**, P < 0.001. **c** Changes in the age-dependent regulation of *gfp* 3'UTR reporter translation upon inhibition of PKA. Ratios of aged:young mean GFP intensities were calculated for control and PKA dominant negative (dn)-expressing brains for each replicate. Mean values obtained for the PKA dominant negative context were then divided for each replicate by those obtained for the control conditions and plotted in a log-2 scale. Three to four replicates were performed for each *gfp*−3'UTR reporter RNA and the distributions of corresponding data points are represented in Supplementary Fig. 9b. Error bars represent s.e.m. Source data are provided as a Source Data file.

are consistent with previous ex vivo work indicating that shared RNP components may establish overlapping interaction networks competing for the recruitment of granule-specific clients[37]. More-over, they illustrate how biological systems can efficiently modulate RNP condensate properties and composition through physiological tuning of RNP component stoichiometry. Notably, increasing the dosage of Me31B through addition of an extra copy of *me31B* in

young flies is not sufficient to trigger aging-like condensation (Supplementary Fig. 10a), suggesting that additional factors contribute to the age-dependent remodeling of neuronal RNP granules. These may include increased levels of other yet to be discovered scaffolds with limiting concentration in young flies, or age-dependent changes in the valency or binding affinities of critical RNP components.

**PKA activity regulates the recruitment of Imp and the translation of its target mRNA *profilin*.** The selective genetic screen we performed led us to uncover that PKA kinase activity is required for the recruitment of Imp to RNP granules, while being largely dispensable for the condensation of other RNP components including Me31B, Tral or HPat1. Although a role for PKA in RNP condensate regulation has previously been described in yeast[74], ectopic activation of PKA prevented P-body assembly in this system, while pharmacological inhibition triggered condensation. Such a regulatory role was shown to be mediated by phosphorylation of the scaffold protein Pat1[74,75], a protein whose *Drosophila* ortholog (HPat1) localizes to neuronal RNP granules (Supplementary Fig. 1d). *Drosophila* HPat1, however, lacks PKA consensus sites ((R/K)(R/K)XS/T) and is still recruited to RNP granules upon PKA inhibition, indicating it is likely not a primary target of PKA in this context. Imp, in contrast, is an attractive candidate target as it is phosphorylated on Serines in *Drosophila* brains (Supplementary Fig. 11a) and contains putative PKA phosphorylation sites. Mutating three of these sites (S58A, S98A and T349A), nonetheless, did not impact on the condensation of endogenous Imp in aged flies (Supplementary Fig. 11b), indicating that PKA acts either through phosphorylation of other sites or through phosphorylation of other targets that remain to be identified. Another interesting question is whether PKA activity itself is modulated upon aging, such that it would promote recruitment of Imp mainly in aged flies. Although previous work had suggested that both expression levels and activity of PKA do not vary in the aging *Drosophila* brain[76], this analysis was performed on entire brain lysates and thus did not address potential population-specific differences. To measure PKA activity specifically in aging MB neurons, we thus expressed the PKA-SPARK GFP biosensor, which generates reversible fluorescent condensates upon PKA-dependent phosphorylation[77,78]. This experiment further confirmed that PKA activity is not higher in aged MB neurons (Supplementary Fig. 10b), and thus that global increase in PKA activity cannot account for the differential recruitment of Imp observed when comparing young and aged brains. Consistent with this idea, over-expressing the catalytic subunit of PKA in young flies did not increase the number of Imp-positive granules (Supplementary Fig. 10c). Given that PKA can be activated very locally by neuronal activity[79], and in light of recent work showing that neuronal activity modulates DDX6 RNP granules in maturing neurons[80], an interesting possibility is still that local age-dependent changes in activity patterns may participate to the observed modifications of neuronal RNP granule properties.

**RNP condensation as a mechanism to regulate the neuronal translatome upon aging.** Aging has long been associated with alterations in gene expression and proteostasis[81]. However, the relative contribution of transcriptional, post-transcriptional and post-translational changes to age-dependent modifications of protein content has only recently started to be explored through the systematic integration of RNA-seq, Ribo-seq and advanced mass-spectrometry analyses. Unexpectedly, analyses performed in rat brain have revealed that the fraction of transcripts and proteins exhibiting significant up- or down-regulation upon aging is in fact relatively small (<10%)[82–85]. These studies also uncovered that changes in the translational output of specific sets of genes could be observed independently of changes at the RNA levels[82], highlighting the importance of post-transcriptional mechanisms in age-dependent regulation of gene expression.

Consistent with these findings, our in vivo analysis of translation reporters did not reveal a general decrease in translation efficiency over time, but rather transcript-specific

responses. While the Imp target *profilin* RNA showed age-dependent recruitment to RNP granules and translation repression, the non-granule associated *SV40* or *camk2* RNAs did not exhibit such a trend. How such specific changes in gene expression relate to the physiology of aging largely remains to be explored, but we have shown that decreased translation of *profilin* mRNA correlates with an age-dependent decrease in Profilin protein levels, as well as with the reorganization of the F-actin cytoskeleton. Tightly regulating Profilin levels might be particularly important for neuronal cells, as high levels of Profilin has been associated with the ectopic formation of Stress Granules[86]. It might however come at a cost in the context of aging, as decrease in Profilin function has been associated with age-related neurodegenerative diseases such as ALS[87,88].

At the molecular level, translational repression could be prevented in contexts where RNP condensation is inhibited (*me31B* RNAi) or the recruitment of Imp to granules is altered (PKA inactivation), suggesting that RNP component condensation may be an important determinant of translational control. Noteworthy, our analysis on *profilin* indicated that only a fraction (~15%, Supplementary Fig. 7c) of RNA molecules is recruited to visible RNP granules in aged brains, a number similar to what has been described in mammalian systems[25,89]. How this then impacts on global protein levels is still unclear, but it may primarily affect low abundance transcripts and/or transcripts exhibiting low translation efficiency. An alternative non-exclusive hypothesis is that recruitment to RNP condensates is underestimated in our analyses based on the detection of RNP granules of sizes above the resolution limit. How RNP granules may inhibit translation also remains to be investigated, but our previous immunostaining experiments indicated that this may be achieved by segregating away the translation machinery. The ribosomal 60 S subunit component RpL32, indeed, could not be found in Imp-positive granules, suggesting that ribosomes are not present in, and thus likely excluded from those granules[48]. Altogether, our results have opened perspectives in the field of aging, raising the interesting hypothesis that targeting of RNA species to RNP granules hosting translational repressors may represent a mechanism employed by neurons to regulate the translation of specific sets of transcripts upon aging.

## Methods

***Drosophila* stocks and genetics**. Fly crosses were performed on standard media and raised at 25 °C unless specified. For aging experiments, flies were transferred to fresh media every 3 days until reaching the age of 35–39 days. For screening of pathways involved in Imp condensation and for the PKA inactivation experiments shown in Figs. 6, 8b and Supplementary Figs. 5 and 6a, adult-specific inactivation was carried out using a tubulin-Gal80ts;;OK107-Gal4 line. Specifically, flies were raised at 18 °C until eclosion, then transferred to 29 °C to allow transgene expression, and aged to 30 days. For analysis of GFP-3' UTR reporter expression in mutant contexts, flies were either raised at 18 °C and switched to 25 °C 1–2 days before eclosion (*me31B* RNAi experiments), or maintained at 25 °C throughout development and adulthood (PKA inactivation experiments with various reporters) and aged to 35–38-days.

The following fly stocks were used in this study: GFP-Imp protein-trap line #G080[60]; Me31B::EGFP and Me31B::mTomato knock-in lines[24]; w; *me31B*Δ1 FRT40A/CyO, w;*me31B*Δ2 FRT40A/CyO and *me31B* gDNA[63]; UAS-*me31B* RNAi (BDSC #33675); UAS-*imp* RNAi (BDSC #34977); UAS-PKA C1 K75A (PKA catalytic dead subunit; BDSC #35557); UAS-PKA C1 RNAi (BDSC#31277); UAS-*amn* RNAi (BDSC #25797 and VDRC #5606); UAS-mcherry-Fascin[71]; UAS-GFP-PKA-SPARK[78]. UASp-EGFP 3'UTR reporter lines (*SV40* 3'UTR, *profilin* 3'UTR and *camkII* 3'UTR) were described in[24] and expressed in MB neurons using OK107 Gal4. The mCherry-Rin, HPat1-mTomato and Me31B-mTomato knock-in lines were generated using the CRISPR/Cas9 technology[90].

**Preparation of *Drosophila* brains for imaging**
*Immunostaining*. Brains were dissected in cold PBS 1X for 1 h and fixed in 4% formaldehyde for 30 min. After fixation, brains were washed thrice in 0.1% PBS/Triton-X (PBT). Brains were then blocked overnight in PBT supplemented with 1% BSA and incubated with the following primary antibodies: rabbit α-Imp (1:1000; Medioni et al., 2014); rat α-Imp (1:1,000; Medioni et al., 2014); rabbit α-Me31B

(1:3,000; gift from C. Lim); mouse α-Me31B (1:3,000; gift from A. Nakamura); rabbit α-HPat1 (1:1,000; gift from A. Nakamura), rabbit α-Tral (1:1,000; gift from A. Nakamura), rat α-Staufen (1:1,000; gift from A. Ephrussi); rabbit α-GFP (1:1,000; Molecular Probes, A-11122), mouse α-Profilin (1:100; DSHB, chi 1 J clone); rabbit α-p62 (1:1,000; gift from Gabor Juhasz), mouse α-Ubiquitin (1:500; Enzo Biosciences Cat # BML-PW8805), rabbit α-PABP (1:1,500; gift from C. Lim). After incubation in primary antibodies, brains were washed thrice in PBT 0.1% and incubated with α-rabbit, α-mouse or α-rat secondary antibodies (1:1,000 dilution) conjugated with Alexa Fluor 568/488/647 for 2 h at room temperature or overnight at 4 °C. Fixed and stained brain samples were mounted in vectashield (Vector Laboratories) medium.

*Detection of endogenous fluorescent signals.* For the detection of endogenous GFP signals, flies were dissected in cold PBS 1X and fixed in 4% formaldehyde for 30 min. Fixed samples were then washed thrice with 0.1% PBT and directly mounted in vectashield (Vector Laboratories) medium.

*Pharmalocgical inhibition of PKA.* Brains were disssected in HL3 buffer (NaCl 70 mM, KCl 5 mM, MgCl$_2$ 4 mM, trehalose 5 mM, sucrose 115 mM, HEPES 5 mM, NaHCO$_3$ 10 mM, pH 7.2–7.3) for 30 min maximum and incubated for 40 min with 50 μM H-89 (PKA inhibitor; Sigma #371962) at room temperature. Brains were then fixed in 4% formaldehyde for 30 min. Fixed samples were then washed thrice with 0.1% PBT and transferred to vectashield (Vector Laboratories) medium.

**Single molecule fluorescent in situ hybridization (smFISH).** *Drosophila* brains were dissected in cold RNase-free PBS. Dissected brains were then fixed in 4% formaldehyde in PBS for 1 h at 4 °C and rinsed twice with PBS. Brains were dehydrated overnight in 70% ethanol and rinsed the day after in wash buffer (10% formamide in 2x SSC) for 5 min at room temperature. Brains were then incubated overnight, at 45 °C, and under agitation, with Quasar®570/670- labeled Stellaris® Probes in 100 μL hybridization buffer (100 mg/mL dextran sulfate, 10% formamide in 2x SSC). *gfp, camk2* and *profilin* probes were used at a final concentration of 0.125 μM, 0.0625 μM and 0.25 μM respectively. After hybridization, brains were washed for 30 min in pre-warmed wash buffer under agitation, at 45 °C. This step was followed by a further 5 min-wash in 2x SSC at room temperature and by mounting in vectashield (Vector Laboratories) medium. Sequences of the probe sets used to detect *profilin, egfp* and *camk2* mRNAs listed in Supplementary Table 2.

**Image acquisition.** Brain samples were imaged using a LSM880 confocal equipped with a airyscan module and a 63 × 1.4 NA oil objective. Images were taken with a 0.04 μm pixel size and were processed with the automatic airyscan processing module of Zen (strength 6.0).

For analysis of EGFP-3'UTR reporters, freshly mounted samples were imaged using a Zeiss LSM780 or 710 confocal microscope equipped with GaAsP detectors and a Plan Apo 63 × 1.4 NA oil objective.

**Fluorescence recovery after photobleaching (FRAP).** GFP-Imp or Me31B-GFP brains were dissected in Schneider's medium and then mounted in polylysinated Lab-Tek chambers. Once properly oriented (dorsal side towards the bottom of the chamber), brains were covered with a gas-permeable membrane and hold in place using a steel ring[91]. Halocarbon oil was added on top of the membrane to avoid evaporation. FRAP experiments were performed on a Nikon microscope coupled with a Yokogawa spinning head and an Andor EM-CCD camera. Imaging was performed using a Plan Apo 100X oil 1.2 NA objective and a 488 nm laser line. The metamorph software was used to acquire images (1 image every 0.24 s) and to bleach the samples. Samples were bleached with a 488 nm laser, using the point laser method. A maximum of 10 granules were bleached per brain.

Fluorescence signals were measured using the following procedure. First, images were aligned using the stack shuffling plugin of ImageJ. Then, to measure granule intensity over time, granule positions were marked manually and an ROI (3 * 3 pixels) was saved for each granule in the ROI manager. ROI mean fluorescence intensities were then calculated using the multimeasure option of ImageJ ROI manager. A double normalization was applied to intensity values, which consisted of bleach correction followed by normalization to pre-bleach intensities.

**Live imaging.** Brains of 5-day old flies were dissected in cold Schneider's medium and then mounted in polylysinated Lab-Tek chambers. Once properly oriented (dorsal side towards the bottom of the chamber), brains were covered with a gas-permeable membrane and hold in place using a steel ring[91]. Halocarbon oil was added on top of the membrane to avoid evaporation. Movies were acquired on an inverted Zeiss LSM880 confocal microscope equipped with an airy scan module and a 63 × 1.4 NA oil objective. Images were acquired every 10 s for 15 min, with a pixel size of 0.028 μm.

**Image analysis**

*RNP granule detection.* ROIs containing 6–7 cells were cropped from single z slices and processed via the following steps: (1) resizing to a factor of 1 using the

Laplacian Pyramid plugin on ImageJ, (2) rescaling to enhance contrast and to keep 0.01% pixels saturated, and (3) converting 32 bit images to 16 bit in order to change float numbers to integer values. Granules were detected using the Small Particle Detection (SPaDe) algorithm (https://raweb.inria.fr/rapportsactivite/RA2016/morpheme/uid13.html)[92]. Cutoff size for granules was set to 4 pixels and thresholds used for detection of Imp granules, Me31B granules, *gfp* and *profilin* RNA were 0.62, 0.42 and 0.22 respectively. For each analyzed field, the number of detected objects, their individual size (in pixels) and their masks were extracted.

*Measurement of intra-granule concentration and partition coefficients.* Partition coefficients were defined as the ratio of the maximal intensities of Me31B or Imp signal in individual RNP granules over the average intensity of the diffuse cytoplasmic signal. Maximal RNP granule intensities were measured with ImageJ on the original raw images, using the masks generated by SPaDe. Average diffuse signals were estimated for each imaged field by manually defining a region of interest devoid of granules and by measuring the mean intensity of Me31B or Imp in this region. Intra-granule concentrations were defined as the mean intensities of Me31B or Imp signal in individual RNP granule masks.

*Colocalization.* Masks of granules generated by SPaDe were converted to binary images using ImageJ. Colocalization was measured with the JACoP plugin of ImageJ, using binarized masks corresponding to different channels and the centroid-Mask method. Ratio of colocalizing spots to total number of spots was calculated. Fold changes were calculated by normalizing the data to the young conditions.

*Reporter quantification.* Except for *me31B* RNAi experiments where single sections were analyzed, maximal intensity projection of Z stacks was performed and 2 ROIs were selected per brain. The mean GFP intensity was calculated for each ROI and normalized to the average of respective controls.

*Total amount of protein.* Two ROIs containing 6–7 cells were selected from each brain and mean intensity calculated for each ROI. Data were normalized to values of respective controls.

*PKA activity.* PKA activity was estimated using the GFP-PKA-SPARK biosensor that undergoes phase separation upon phosphorylation by PKA. PKA activity was equated as the ratio of the total amount of phase-separated GFP (condensate mean intensity X area) to the total intensity of the diffuse cytoplasmic signal excluding phase-separated GFP. Masks of phase-separated GFP + condensates and cytoplasm without phase-separated GFP were generated automatically using the SPADE algorithm. Total intensities were measured using Fiji macros on masks generated by SPADE.

**Solubility assay.** 150 μL of age-matched GFP-Imp fly heads were collected and homogenized in 500 μL of RIPA supplemented with protease and phosphatase inhibitors (lysis buffer). Lysates were incubated under agitation for one hour at 4 °C and then centrifuged at 1,000 g for 10 min at 4 °C. 40 μL of supernatant were kept aside and defined as input sample and the rest was complemented with lysis buffer to reach 500 μL. Fractions were then subjected to ultra-centrifugation for 45 min at 100,000 g, 4 °C. Supernatants were collected (soluble fraction), and pellets were dissolved in 60 μL Urea buffer (9 M urea, 50 mM Tris-HCl pH8, 1% CHAPS) to collect insoluble fractions. 5% of each fraction were subjected to electrophoresis and then blotted to PVDF membranes. Membranes were then blocked for 1 h at room temperature and incubated overnight with primary antibodies. The following primary antibodies were used: rabbit anti-GFP (1:2,000; Torrey Pines, #TP401) and mouse anti-Tubulin (1:10,000; Sigma #T9026).

**Immuno-precipitations.** Fly heads were snap frozen in liquid nitrogen and lysed using micro pestles in RIPA buffer (0.1% sodium deoxycholate, 1% triton ×100, 0.1% SDS, 150 mM NaCl, 50 mM TRIS pH 7.00) supplemented with Halt™ Protease Inhibitor Cocktail 1:100 (Thermofisher, #78429). Respectively 200 μL and 300 μL of young and old heads were lysed into 500 μL of RIPA buffer. Lysates were incubated under agitation at 4 °C for 30 min, then centrifuged at 400 g for 10 min at 4 °C to remove tissue debris. Supernatants were collected and incubated with 40 μL of equilibrated ChromoTek GFP-Trap® beads (ChromoTek, gt-10, #70112001 A) for 2 h at 4 °C. Beads were washed three times 30 min in RIPA buffer, resuspended in 25 μl 2X SDS loading buffer, and incubated at 95 °C for 5 min for elution and denaturation.

Input and bound protein fractions were subjected to electrophoresis and blotted to PVDF membrane. Membranes were then blocked with 4% BSA for 1 h at room temperature prior to addition of primary antibodies. The following primary antibodies were used: rabbit anti-GFP (1:2,500; Torrey Pines, #TP401); rabbit anti-phospho serine (3ug/mL; Abcam #ab9332) and mouse anti-phospho serine (1:200; Sigma #P3430).

**RT-QPCR.** RNA was isolated from fly head lysates using Trizol (Invitrogen) and used as template for reverse-transcription reaction performed with Superscript III (Invitrogen) and Oligo(dT). 1% of the RT product was then PCR-amplified through QPCR, using the following couples of optimized primers: *qPCRrpl7_*fwd:

5'-CGTGCGGGAGCTGATCTAC-3'/ *qPCRrpl7*_rev: 5'-GCGCTGGCGGTTATG CT-3'; *rp49*_fwd: 5'-CTTCATCCGCCACCA-3' / *rp49*_rev: 5'-CTTCATCCGCC ACCA-3'; *profilin*_fwd: 5'-CTGCATGAAGACAACACAAGC-3' / *profilin*_rev: 5'-CAAGTTTCTCTACCACGGAAGC-3'.

**Statistical analysis and reproducibility**. All data were plotted and statistically analyzed using Graphpad Prism 8. The statistical tests performed for comparison of conditions are all mentioned in the corresponding figure legends. Unless specified in the Figure legend, three independent replicates were performed for each experiment. When possible, the results of individual replicate were plotted using SuperPlots[93].

**Reporting summary**. Further information on research design is available in the Nature Research Reporting Summary linked to this article.

## Data availability

The data supporting the findings of this study are available from the corresponding authors upon reasonable request. Source data for the figures and supplementary figures are provided as a Source Data file.

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

## Acknowledgements

This study was supported by the CNRS, as well as grants from the ANR (ANR-15-CE12-0016 and ANR-20-CE16-0010) and the Fondation pour la Recherche Médicale (Equipe FRM; grant #DEQ20180339161) to F.B. K.P. was supported by a fellowship from the LABEX SIGNALIFE program (#ANR − 11 − LABX − 0028 − 01) and a one year- La Ligue contre le cancer fellowship. Part of this work was also supported by the Joint Usage/Research Center for Developmental Medicine, IMEG, Kumamoto University. We thank the iBV PRISM Imaging facility for use of their microscopes and support (especially B. Monterroso), and L. Palin for excellent technical assistance. We are grateful to members of the Besse group for discussion and advice, in particular to F. De Graeve for help with the SPaDE algorithm. We thank A. Hübstenberger, M. Kiebler, K. Bauer, F. De Graeve and C. Medioni for critical reading of the manuscript, and N. Formicola and E. Aubry for help with image analysis. We are grateful to the Bloomington Drosophila Stock Center and the Developmental Studies Hybridoma Bank for reagents.

## Author contributions

K.P. performed all experiments and quantifications presented in the manuscript, except those presented in Fig. 8a, c and Supplementary Figs. 1k, l, 6a–d, 8c–e, 9 and 10b, which were performed by M.S. A.N. generated the mcherry-Rin, HPat1-mTomato and Me31B-mTomato lines used in this study and produced the α-Me31B and α-Tral antibodies. K.P., M.S. and F.B. contributed to hypothesis development, experimental design and data interpretation. F.B. provided the overall supervision, the funding and wrote the article. All authors discussed the data and commented on the manuscript.

## Competing interests

The authors declare no competing interests.
