## [Peer Review File · Nature Communications]

Title: We would like to coordinate the publication of our manuscript with that of the co-submitted manuscript from the Kiebler group. We have prepared a joint cover image that will be submitted by the Kiebler group.REVIEWER COMMENTS

Reviewer #1 (Remarks to the Author):

In this manuscript, the authors reported novel observation on that RNP components (such as Me31B and Imp) can form large - dynamic granules in aged Drosophila brain. The age-dependent clustering of the RNP components is caused by an increase expression level of Me31B, and this process requires PKA kinase activity. The authors also showed that that profilin mRNAs recruited to RNP condensates upon aging are translationally repressed.

The manuscript represents interesting quantitative imaging works with nicely presented data. However, the functional role of the RNP condensates in aging is understudied. The mechanism link between PKA pathway and the regulation of RNP granule formation remains to be identified. Some key biochemical control experiments need to be added. Please see my specific points below.

1. How the authors interpret the results in Fig S1A, it seems do not show the same “gradual and continuous” clustering change as in Fig 1 O.
2. Fig 2, does Me31B granules colocalized with p62 and Ubiquitin or not?
3. Can the authors report on whether Me31B and Imp (in young and aged brain) are in soluble or insoluble form by Western Blot? It is necessary to show biochemical evidences to rule out the possibility that the granules observed are (partially) in protein aggregate form, especially the Imp granules showed a relatively low recovery rate of 40% in the FRAP experiments.
4. In Fig 4 D, there is a small (but significant) decrease in the number of Imp+ granules in the me31B RNAi strain. Maybe it is not enough to draw the conclusion of “ inactivating me31B prevented the assembly of Imp-containing granules” based on such a small change.
5. It will be necessary to show Western Blot results on the age-dependent increase in Me31B levels, since this is one of the key results of the manuscript.
6. Can the authors test whether the decreased number of Imp+ granules in PKA knockdown background is due to a decrease expression level of Imp by Western Blot?
7. Can the authors perform the translation experiment in the imp RNAi background? This can provide evidence on whether Imp expression level is involved in the translational repression observed.
8. To establish a functional link between the RNP condensates with the translational repression observed, it is important to show that by only blocking the RNP condensates formation (e.g. via point mutants) without changing their expression levels can prevent the translational repression effect.

Minor points

1. On page 4, the authors mention that “We uncovered that cytoplasmic RNP components progressively condense into large non-pathological granules upon aging...”. Can the authors explain what did they based on to draw the conclusion that the granules are non-pathological?
2. The first letter of some Y axis labels are uppercase, some are not. Recommend to checked the guidelines.
3. Can the authors discuss the potential functions of the specific translational repression of proteins (e.g.

profilin) on aging process?

4. On page 12, "... an extra copy of me31B in young flies is not sufficient to trigger "aging-like" condensation (K.P and F.B., data not shown)", please check the journal's guidelines to see if this is allowed.

Reviewer #2 (Remarks to the Author):

Report on manuscript "RNP components cluster into repressive condensates in the aging brain".

In this manuscript, the authors investigate how aging affects the formation of RNP granules in the Mushroom Body of the *Drosophila* brain. The authors reports that DDX6 (a P-body marker) and IMP granules are affected by aging: IMP becomes recruited to DDX6 granules in old neurons and DDX6 granules become larger. The increase of DDX6 granule size are due to increased expression of the protein, while IMP recruitment to DDX6 granules is affected by the protein kinase A (PKA) pathway. However, IMP does not appear to be a direct target of PKA. The authors then investigate the metabolism of profilin mRNA, a direct target of IMP. The authors show that more mRNA is found in IMP granule in old brains and that the mRNA is less translated, while PKA inhibition reverts both granule formation and translation repression in old neurons. In conclusion, the authors propose that aging changes mRNA expression via condensate formation.

The finding that IMP granules (mainly) and DDX6 granules (to a lesser extend) are affected by aging in the *drosophila* brain is interesting, as is the finding that this correlates with the granule localization and expression of an IMP mRNA target. However, I believe that the study needs more mechanistic and functional insights to make a full story.

First, it is unclear how PKA acts on IMP to affects its localization to DDX6 granules, and this is a key issue to understand the possible mechanisms and to fully interpret the effects of PKA inhibition. Is PKA repressing the translation of some mRNAs ? Is it acting on DDX6 granules ? on an IMP partner ?

Second, is PKA activity different in the old vs young neurons ? in other words, are the experiments that manipulate PKA reflecting something that happens in vivo ?

Third, it is difficult to conclude from the data whether the recruitment of profilin mRNA to granule in old brain is a cause or a consequence of translational repression.

Finally, the consequences of all this on the function of the young or old neurons is unclear. Are the levels of endogenous profilin protein different in young and old neurons ? Is a 25%-50% reduction in the level of profilin sufficient to affect the function of neurons and to contribute to aging phenotypes ?

The significance of the observations reported should thus be defined in more depth and the proposition that aging changes mRNA expression via condensate formation should be supported with additional data.

Other Comments:

-What happens to young neurons if PKA is over-activated ?

-What happens to neuronal processes ? Is it similar than in the cell body (more IMP granules in old neurons) ?

-Are these effects specific to the neurons analyzed or can they be seen in other neurons (or in a brain extract) ?

-In Figure 7, the number of mRNA in granules should be counted using DDX6 as a marker, not IMP, as IMP recruitment to granules is very low in young neurons. The fraction of endogenous and reporter mRNA in granules should also be calculated.

-For the quantification of GFP intensities, it may be good to back them using Western blots or other.

Reviewer #3 (Remarks to the Author):

This manuscript focuses on the formation of neuronal granules by the conserved DEAD-box helicase, Me31B in response to aging. Interestingly, Me31B levels increase in aging brains, resulting in fewer but larger (amount of Me31B per granule?) RNP granules and this increase depends on PKA. Me31B granules in aged brains more frequently contain Imp protein than Me31B granules in young brains suggesting that Me31B is limiting for the recruitment of client proteins like Imp. Although necessary, increasing Me31B levels isn't sufficient to form larger granules and promote recruitment of Imp, however, suggesting that there is some other feature of aging that is required that is not addressed here. The authors refer to these Me31B and Imp containing granules as poorly diversified, although this is an overstatement given that they only examine one client protein. Increased association of Imp with Me31B correlates with a decrease in the level of translational reporter for an Imp target RNA, profilin, suggesting that the age-dependent incorporation of Imp into these granules is a mechanism to repress its translation.

My main concern is that the key finding that relates to aging is that Me31B levels increase but nothing in the manuscript addresses how this happens. Instead the authors focus on the idea that granules are being remodeled and changing their functions, but there is little evidence for any remodeling of change in function going on. Furthermore, perhaps this is a semantic issue, but the way that the manuscript is

written seems to suggest that Me31B is acting to assemble different types of granules in young versus aged brains. A simpler interpretation is that Me31B assembles one type of granule that recruits varying amounts of Imp from the cytoplasmic pool, depending on the amount of Me31B. Finally, there is no evidence that the function of the Me31B granules actually changes during the aging process – they may always be repressive for Imp translation, there is just a shift from the amount of Imp in the cytoplasmic translating pool relative to the amount in Me31B granules in the young versus aged brain.

Specific Issues

- 1) It isn't clear how the authors are defining granule size (large vs small) – are they referring to the measured size of the granule or the fluorescence intensity (which would relate to number of molecules)? This should be clarified.
- 2) From the images shown, it is not so clear that there are more Imp particles in aged versus young brains but rather there are fewer, brighter spots in aged brains.
- 3) Colocalization of profilin RNA and Imp seems very infrequent and there is no quantification. Is there a quantifiable change in granule association of profilin in aged versus young brains?
- 4) Based on FRAP experiments, Me31B containing granules seem to be more liquid-like and Imp less liquid-like based on FRAP experiments and the properties don't seem to change between young and aged brains. However, in aged brains, most Me31B also contain Imp. When the authors performed FRAP experiments to analyze Imp in aged brains, were these only performed on Me31B- Imp+ granules or were these performed on Me31B+ Imp+ granules? If the latter (or a mixture), then it would seem that Me31B is not determining granule properties since Imp granules remain less liquid-like in aged brains.
- 5) I am not convinced by the claim that there is a change in the sorting of RNP components to distinct granules or that Me31B and Imp condense into large multiphase granules in aged brains. Imp does associate with Me31B granules in young brains and the increase in the Me31B+ Imp+ granules from young to old is only several fold. Do the Me31B granules in young brains that contain Imp have more Me31B than the granules that don't contain Imp? It seems equally possible that as the amount of Me31B increases with aging, more Imp is recruited from the cytoplasmic pool into Me31B granules, which then just increases the frequency of finding Imp in Me31B granules. Related to this and the above point, how does the frequency of Me31B- Imp+ granules change? Do all Imp+ granules contain Me31B in aged brains?
- 6) Was the experiment in Figure 4 performed using young or aged brains? Is Me31B required for granules to form in young brains or just for the larger granules in aged brains? This isn't shown and should be made clear.

In addition, it isn't clear why the authors conclude that inactivation of Me31B prevents assembly of Imp-containing granules – rather it seems to prevent the formation of Me31B granules that recruit (see point

#5).

7) In Figure 5, when Me31B is reduced, it seems like colocalization of Imp with Me31B is not affected – does the level of Me31B still show an increase in aged brains? Furthermore, it is difficult to compare the data in Figures 5D and 5E, which is necessary to be able to interpret the effect of reducing the level of Me31B on Imp. A cumulative frequency distribution for Imp should be shown and, vice versa, the number of granules plotted for Me31B similarly to E.

Also related to the experiment in Figure 5, the conclusion that "MB gamma neurons tune the concentration of limiting scaffold protein Me31B on aging so as to trigger condensation into large granules. This only mildly promoted recruitment of Imp to Me31B granules..." is contradictory to data presented earlier showing that nearly all Me31B granules contain Imp in aged brains.

8) Rather than PKA preventing condensation of Imp into larger granules it seems equally plausible that PKA prevents the incorporation of Imp into Me31B granules.

9) Figure 7C is not referred to in the text. Perhaps the image is supposed to illustrate colocalization of profilin with Imp quantified in Figure 7A but there doesn't seem to be much evidence for colocalization based on the image.

10) Is the gfp-profilin 3'UTR also localized to granules – this should be shown, especially because the image in Figure 7C is not convincing.

11) Why rely on a reporter for profilin translation rather than detecting Profilin levels directly? Additionally, how can the authors distinguish whether the change in translation is due to granule association versus just an effect of age? Performing the experiment in an me31B knockdown where the granules don't form would be more informative. Although the PKA experiment is an attempt to do this, PKA could have other effects on translational activity independent of granules.

Minor points

1) The word "cluster" seems like an odd choice to refer to the accumulation of RNP components in granules.

2) How do the authors know that the granules they observed in aged brains aren't pathological?

3) Why is it important to repress profilin translation in aged brains? The authors should comment on this in the Discussion.

Reviewer #4 (Remarks to the Author):

The manuscript by the Besse group examines the dynamics of RNP granules in the physiological context of the aging *Drosophila* brain. The study first compares known granule markers (Imp and Me31B, among others) between young and aged *Drosophila* brains and describes a specific enrichment of these markers in granules upon aging. Next, the authors provide evidence that aging-associated granules are dynamic and result from the coalescence of smaller pre-existing granules. The authors then link these granules to an increase in Me31B dosage observed during normal aging and suggest they are sites of age-dependent localized translational repression. This repression is sensitive to PKA inactivation, which also differentially reduces Imp (strongly) and Me31 (mildly) recruitment to granules, providing a link between RNP component recruitment and translational repression in condensates.

This manuscript documents for the first time, to my knowledge, the condensation of RBPs in granules in the context of normal aging. I find this interesting and promising in which it adds novel physiological depths to the field. However, the manuscript falls short in exploring the functional consequences of this condensation. In particular, the GFP-3'UTR reporter is a poor choice to monitor localized translation (especially since alternative, more recent, systems exist: for example, the SunTag system recently used by the authors PMID: 33890854). Moreover, experiments based on PKA inactivation provide weak functional evidence due to the many non-specific effects of such inactivation.

Major comments

Condensation characterisation:

1. In Fig. S1, the authors claim that certain RBPs (Tral, HPat1, Stau) cluster upon aging while others do not (PABP and Rin). This conclusion needs to be supported with quantitative measurements.
2. The authors set out to investigate if the Me31B+ Imp+ RNP granules observed upon aging result from the merging of two types (Me31B+ Imp+ or Me31B+ Imp-) of pre-existing granules. To address this, they quantify the fraction of Me31B + Imp+ granules to Me31B + Imp- in aged and young flies and deduce that an increase in ratio must result from changes in the trafficking of granule RNP components upon aging. However, this observation can simply be the result from new diffuse Imp localizing in granules rather than two distinct types of granules coalescing. Live imaging would provide a stronger argument for coalescence. However, there is no such coalescence event visible in video 1.
3. In Fig. 3, can the authors please also provide the proportion of Me31B+ Imp+ and Me31B+ Imp- granules in young and aged flies rather than just the aged:young ratio? This will improve data clarity and readability.

Me31B dosage study:

4. The authors claim that RNAi-mediated knockdown of me31B but not Imp prevents the assembly of granules. Please provide the same granule number quantification for the Imp RNAi experiment to

support this claim.

5. To prove that Me31B dosage affects its condensation, the authors remove a copy of me31B and observe that Me31B granules in aged flies resemble the young phenotype. The reduction of Me31B granule size can however be due to the lower levels of Me31B expression rather than a change in compartmentalization (i.e. less cytoplasmic Me31B available overall, and not a change in Me31B's localization). Comparing the partition coefficient of Me31B between WT and the two deletion lines would be informative in this case.

6. Please provide quantitative evidence in this manuscript that removing a copy of the Me31B gene reduced stable levels of the protein.

GFP translation reporter

7. The authors use a GFP reporter fused to the 3'UTR of the granule enriched RNA profilin or two other control 3'UTRs (those of SV40 and the non-granule enriched transcript camk2) as a tool to measure the translational regulation in granules. This approach to study localized translation is questionable and has several issues: (i) the GFP- profilin 3' UTR construct was not shown to localize in granules , (ii) total GFP fluorescence is not an indicator of translational output per se, but rather protein steady state levels that can be linked to other processes such as protein stability or turnover (even if RNA levels revealed by smFISH remain similar), (iii) even if translation inhibition was taking place upon aging, it may not be exclusive to granule RNAs, as suppression of non-granule localized GFP transcripts in the cytosol will also contribute to the total reduction in fluorescence.

8. To directly demonstrate localized translational repression in granules, the authors should take advantage of the SunTag system recently used by them in the same organism.

Role of PKA

9. The authors notice a translational de-repression upon inactivating PKA. Since PKA repression was associated with a reduction of RBP recruitment in granules (significant for Imp but mild for Me31B), the authors suggest that granule-localized Imp must be critical for translational regulation there. In addition to having the same issues of using GFP steady state fluorescence to monitor translation, the approach of inhibiting PKA is not specific enough and can alter many cellular pathways. Thus, the observed effects on GFP fluorescence levels cannot be solely attributed to lower levels of Imp in granules. For instance, Me31B is also mildly affected by the PKA inactivation.

Minor comments

1. In the third paragraph of the introduction, the authors state that RNP condensates have up to dozens of RNA and protein molecules. However, recent transcriptomic and proteomic approaches (that the authors cite) suggest a much higher number. Please clarify this sentence.

2. It would be good if the authors can better explain how partition coefficients are calculated in the material and methods section since they are essential to several conclusions of the manuscript.

3. In Fig. 1L, Does the young condition have only one replicate? Is this the cumulative percentage curve?

4. Live imaging description in the methods section states that the brains of 5-day old flies are imaged, whereas the legend of video 1 states that it was 10-day old flies.

We thank our referees for their thorough analysis of our article, and for their comments and questions, which helped us improve the clarity and the quality of our manuscript. In the revised version, we now better describe the causal link between condensation and translational repression, showing that translational repression is not observed when condensation is inhibited. Furthermore, we clarified the role of PKA and the specificity of the phenotypes associated with its loss of function by providing evidence that i- both chronic and acute inhibition of PKA inhibit the recruitment of Imp to RNP granules, ii- PKA inhibition does not inhibit the recruitment of other RNP components tested, and iii- PKA inhibition does not induce a general increase in translation but rather specifically de-represses the translation of the granule-associated *profilin* RNA species.

These new results, together with others described below, and with the important re-organization of the main text we have performed, have resulted in an extensively modified version, where we better describe and characterize the remodeling of RNP granules occurring upon aging and address most of the points raised by the referees. One main Figure, four supplementary Figures and several Figure panels were added, resulting in a manuscript containing 8 main Figures and 11 supplementary ones.

Reviewer #1 (Remarks to the Author):

In this manuscript, the authors reported novel observation on that RNP components (such as Me31B and Imp) can form large - dynamic granules in aged *Drosophila* brain. The age-dependent clustering of the RNP components is caused by an increase expression level of Me31B, and this process requires PKA kinase activity. The authors also showed that that profilin mRNAs recruited to RNP condensates upon aging are translationally repressed.

The manuscript represents interesting quantitative imaging works with nicely presented data. However, the functional role of the RNP condensates in aging is understudied. The mechanism link between PKA pathway and the regulation of RNP granule formation remains to be identified. Some key biochemical control experiments need to be added. Please see my specific points below.

1. How the authors interpret the results in Fig S1A, it seems do not show the same “gradual and continuous” clustering change as in Fig 1 O.

The strong increase in Imp+ granule number observed when comparing 2 day- and 10 day-old brains likely has technical, rather than biological causes. As Imp exhibits a strong diffuse localization in the cytoplasm of young flies, detection of low intensity granules over such a background is sub-optimal at this age. By facilitating the detection of low-intensity granules, the slight decrease in Imp cytoplasmic signal observed at 10 days will thus have a threshold effect, thereby abruptly impacting on granule number.

2. Fig 2, does Me31B granules colocalized with p62 and Ubiquitin or not?

In aged brains, Me31B and Imp co-localize in the same large granules, so the expectation was that Me31B, similarly to Imp, would not colocalize with p62 or Ubiquitin. To demonstrate this point, we have performed new co-localization experiments, using Me31B as a marker. As shown in revised Figure 2A and 2B, neither p62 nor Ubiquitin accumulates in the large Me31B-positive granules found in aged brains.

3. Can the authors report on whether Me31B and Imp (in young and aged brain) are in soluble or insoluble form by Western Blot? It is necessary to show biochemical evidences to rule out the possibility that the granules observed are (partially) in protein aggregate form, especially the Imp granules showed a relatively low recovery rate of 40% in the FRAP experiments.

We thank the referee for his/her suggestion. To address this point, we prepared brain lysates from young and aged flies and performed differential centrifugation to separately collect soluble and insoluble fractions. As shown in the revised Supplementary Figure 2C, both Imp and Me31B are mostly found in the soluble fraction. Furthermore, no increase in the amount of insoluble proteins was observed upon aging. Thus, while we cannot exclude that a small fraction of granules is in protein aggregate form, this fraction does not increase with age, further confirming that the large granules observed in aged brains do not result from aggregation. This finding does not contradict the data from FRAP experiments, as i- a 40% recovery is observed within seconds, indicating that the Imp molecules are far from being static and ii-it is well described that the various components of liquid-like condensates can exhibit low and high turnover rates.

4. In Fig 4 D, there is a small (but significant) decrease in the number of Imp+ granules in the me31B RNAi strain. Maybe it is not enough to draw the conclusion of “ inactivating me31B prevented the assembly of Imp-containing granules” based on such a small change.

We agree with the referee that our conclusion was probably too strong and modified it. In the revised version of the manuscript, these data are presented in revised Supplementary Figure S3B-D.

5. It will be necessary to show Western Blot results on the age-dependent increase in Me31B levels, since this is one of the key results of the manuscript.

We agree with the referee that validating the age-dependent increase in Me31B levels by independent means was important. However, the sensitivity of Western Blot analyses is low in our experimental model, as protein lysates can only be performed with extracts from entire brains, which contain a high diversity of cell-types expressing Me31B at different levels. As cell-specific behaviors are not to be excluded, we used a different approach to validate the age-dependent increase in Me31B levels initially obtained by immunofluorescence using anti-Me31B antibodies. Specifically, we directly quantified the GFP signals produced in the Me31B-GFP knock-in line we generated using the CRISPR/Cas9 methodology. As shown in the revised Figure 4B, a very similar increase in Me31B levels was observed in the Me31B-GFP line, confirming age-dependent increase in Me31B levels.

6. Can the authors test whether the decreased number of Imp+ granules in PKA knockdown background is due to a decrease expression level of Imp by Western Blot?

To address this question, and because of the limits of Western Blots (see previous point), we quantified and compared GFP-Imp signals in control aged flies and aged flies expressing the PKA dominant negative construct. As shown in our revised Supplementary Figure 5C, no significant difference in the level of Imp protein could be observed.

7. Can the authors perform the translation experiment in the imp RNAi background? This can provide evidence on whether Imp expression level is involved in the translational repression observed.

The experiments shown in revised Figure 4B indicated that Imp levels are not significantly changing upon aging. To however assess whether Imp is involved in the observed translation repression, we quantified the signal produced by the GFP-*profilin* 3'UTR reporter upon *imp* RNAi. This revealed a decrease, rather than an increase, in the intensity of GFP signal when compared to controls (Figure A).

As such a decrease was also accompanied by a decrease in the total amount of *gfp-profilin* 3'UTR RNA molecules (not shown), this indicated that Imp probably regulates the stability of the reporter RNAs and prevented us from estimating *imp*-dependent translation repression.

Figure A. Expression of the GFP-*profilin* 3'UTR reporter upon *imp* RNAi. GFP signal intensities were normalized to the control condition. Two replicates were performed and data plotted with different colors. **, $P < 0.01$ (unpaired t-test)

8. To establish a functional link between the RNP condensates with the translational repression observed, it is important to show that by only blocking the RNP condensates formation (e.g. via point mutants) without changing their expression levels can prevent the translational repression effect.

We agree with the referee that having a point mutant that would i- be compatible with *in vivo* analyses of adult brains (*i.e.* viable until adulthood) and ii- exclusively block RNP condensation would be very nice. To our knowledge, however, no such mutant has been identified so far. To strengthen the model that age-dependent condensation plays an important role in translational repression, we inhibited RNP condensation by inactivating *me31B* through RNAi, and analyzed GFP-*profilin* 3'UTR in young and aged flies. Interestingly, these experiments revealed i- that Me31B is not involved in translational repression of *profilin* RNA in young flies (revised Supplementary Figure S9A), and ii- that no decrease in GFP-*profilin* 3'UTR expression was observed in this condition upon aging (revised Figure 8A). This new result thus indicates that RNP condensation is an important contributor to age-dependent translation repression.

Minor points

1. On page 4, the authors mention that “We uncovered that cytoplasmic RNP components progressively condense into large non-pathological granules upon aging...”. Can the authors explain what did they based on to draw the conclusion that the granules are non-pathological?

The referee is right in pointing out that the pathogenicity of aged granules was not tested. We thus modified the sentence in our revised introduction.

2. The first letter of some Y axis labels are uppercase, some are not. Recommend to checked the guidelines.

We thank the referee for pointing this out. We more carefully checked the guidelines and modified our Figures so that labeling of axes are in “sentence case”, with the exception of labels starting with a transcript name.

3. Can the authors discuss the potential functions of the specific translational repression of proteins (e.g. profilin) on aging process?

Why specific classes of transcripts would undergo specific translation repression upon aging is indeed interesting and raises the question of the link to the physiology of neuronal aging. To address this question, and as Profilin is a known inducer of F-actin polymerization, we compared the levels and distribution of the F-actin binder Fascin in young and aged MB neurons. These experiments revealed that the levels of cortical Fascin decrease upon aging (revised Supplementary Figure S8C-E), indicating age-dependent alterations of the F-actin network. These results, together with hypothesis related to the importance of regulating Profilin levels, are now discussed in the revised version of the discussion.

4. On page 12, “... an extra copy of me31B in young flies is not sufficient to trigger “aging-like” condensation (K.P and F.B., data not shown)”, please check the journal’s guidelines to see if this is allowed.

We now show the corresponding data in the revised Supplementary Figure S10A.

Reviewer #2 (Remarks to the Author):

Report on manuscript "RNP components cluster into repressive condensates in the aging brain".

In this manuscript, the authors investigate how aging affects the formation of RNP granules in the Mushroom Body of the *Drosophila* brain. The authors report that DDX6 (a P-body marker) and IMP granules are affected by aging: IMP becomes recruited to DDX6 granules in old neurons and DDX6 granules become larger. The increase of DDX6 granule size are due to increased expression of the protein, while IMP recruitment to DDX6 granules is affected by the protein kinase A (PKA) pathway. However, IMP does not appear to be a direct target of PKA. The authors then investigate the metabolism of *profilin* mRNA, a direct target of IMP. The authors show that more mRNA is found in IMP granule in old brains and that the mRNA is less translated, while PKA inhibition reverts both granule formation and translation repression in old neurons. In conclusion, the authors propose that aging changes mRNA expression via condensate formation.

The finding that IMP granules (mainly) and DDX6 granules (to a lesser extend) are affected by aging in the *Drosophila* brain is interesting, as is the finding that this correlates with the granule localization and expression of an IMP mRNA target. However, I believe that the study needs more mechanistic and functional insights to make a full story.

1- First, it is unclear how PKA acts on IMP to affects its localization to DDX6 granules, and this is a key issue to understand the possible mechanisms and to fully interpret the effects of PKA inhibition.

Is PKA repressing the translation of some mRNAs ?

We thank the referee for his/her suggestion. To demonstrate the specificity of *profilin* translational de-repression in response to PKA inhibition, we analyzed the translation of non granule-associated RNA

species (*SV40* and *camk2* reporter RNAs) upon expression of the PKA kinase dead variant. These analyses revealed that PKA inhibition does not similarly increase the translation of *SV40* and *camk2* reporters (Figure 8C and Supplementary Figure S9B), and thus that it does not in general boost translation efficiency.

Is it acting on DDX6 granules ? on an IMP partner ?

To get a more comprehensive view on the behavior of RNP components in response to PKA inactivation, we analyzed the distribution of two other granule markers: Tral and Hpat1 upon expression of the PKA kinase dead variant. Interestingly, the behavior of both Tral and Hpat1 was not significantly impacted by PKA inactivation (revised Supplementary Figure S6A,B), indicating that PKA is not regulating granule properties in general, but rather the behavior of specific components. Identifying PKA target(s) and validating it(them) *in vivo* was however not possible in the time frame of revision.

2- Second, is PKA activity different in the old vs young neurons ? in other words, are the experiments that manipulate PKA reflecting something that happens *in vivo* ?

We believe that the functional *in vivo* approaches we have taken, in which we disrupted the activity of endogenous PKA by different means (RNAi, kinase dead protein) in adult fly brains, undoubtedly reflects the role of PKA in controlling RNP component behavior *in vivo*. Whether PKA activity changes upon aging, and whether such changes may underlie the observed age-dependent remodeling of RNP granules, is something that we initially did not address.

To measure the activity of PKA in young and aged fly brains, we expressed in MB neurons the PKA-SPARK GFP biosensor, which generates reversible fluorescent condensates upon PKA-dependent phosphorylation (Zhang et al. 2018; Sears and Broadie 2020). These experiments revealed that PKA activity slightly decreases upon aging (revised Supplementary Figure S10B), a change that cannot account for the difference in behavior observed in young vs aged brains, as genetical inactivation of PKA rather reverts granule component condensation. This is now mentioned in our revised discussion.

3- Third, it is difficult to conclude from the data whether the recruitment of profilin mRNA to granule in old brain is a cause or a consequence of translational repression.

This is indeed an interesting point, and we have performed an important new experiment to address the link between *profilin* translational repression and recruitment to granules. Specifically, we analyzed the expression of the GFP-*profilin* 3'UTR reporter upon RNAi-inactivation of *me31B*, a context in which condensation of Imp is inhibited (revised Supplementary Figure S3A-D). Remarkably, inactivation of *me31B* did not impact on the expression of GFP-*profilin* 3'UTR in young brains (revised Supplementary Figure S9A). However, it prevented age-dependent translational repression (revised Figure 8A), thus suggesting that condensation is required for translational repression.

4- Finally, the consequences of all this on the function of the young or old neurons is unclear. Are the levels of endogenous profilin protein different in young and old neurons ? Is a 25%-50% reduction in the level of profilin sufficient to affect the function of neurons and to contribute to aging phenotypes ?

The significance of the observations reported should thus be defined in more depth and the proposition that aging changes mRNA expression via condensate formation should be supported with additional data.

As shown in our previous Supplementary Figure 5B (and now revised Supplementary Figure S8A), a significant decrease in the levels of endogenous Profilin protein is observed upon aging in MB neurons. As Profilin is a known inducer of F-actin polymerization, we have tested if this drop in Profilin level impacted on F-actin. As displayed in our revised Supplementary Figure S8C-E, we were able to indeed demonstrate that the levels of cortical F-actin decreases upon aging. This new result is consistent with previous observations reporting a decrease in F-actin in neurons upon aging, and with a model in which age-dependent post-transcriptional repression of *profilin* may contribute to age-dependent changes in the properties of the neuronal actin cytoskeleton. This is now discussed in our revised version of the discussion.

Other Comments:

-What happens to young neurons if PKA is over-activated ?

As illustrated in our revised Supplementary Figure S10C, over-expressing PKA in MB neurons of young

flies did not induce premature Imp condensation.

-What happens to neuronal processes? Is it similar than in the cell body (more IMP granules in old neurons) ?

In vivo, MB γ neurons extend processes that fasciculate to form compact bundles composed of more than 600 intermingled axonal branches projecting to the ventral side of the fly brain. As estimating the number of Imp granules in these highly compact structures is challenging, we had to adapt our detection algorithm to quantitatively compare granule number in the axons of young and aged flies. No significant difference could be observed regarding the density (number per surface area) of Imp granules in this condition (Figure B, below). This might reflect that the properties and composition of somatic and process-localized RNP granules are different, as previously reported (Hillebrand et al. 2010). As we cannot exclude that the high density of axonal granules, together with the complexity of images, might prevent accurate comparison of the two conditions, we would prefer not to mention this result in our manuscript.

Figure B. Imp RNP granules in axons. (A,B) Distribution of GFP-Imp signal in MB axon bundles of young (A) and aged (B) brains. The imaged fields correspond to regions defined at the tip of MB medial lobes. (C) No significant changes in the number of Imp RNP granules could be observed. Scale bar: 5 μ m.

-Are these effects specific to the neurons analyzed or can they be seen in other neurons (or in a brain extract) ?

This is an interesting question that we tried to address by analyzing the subcellular localization of Imp in other neuronal populations, focusing on populations where Imp is detectable. While we feel that Imp tends to also condensate upon aging in other neurons (see Figure C; neurons located in the optic lobe), quantifying this phenomenon is challenging, as granules are not as visible as in Mushroom Body neurons. Thus, we would prefer not to include these data in the revised manuscript.

Figure C. Imp RNP granules in non-MB neurons. Note that condensation of Imp appears to occur also in non-MB neurons, although this is less pronounced than in MB neurons.

-In Figure 7, the number of mRNA in granules should be counted using DDX6 as a marker, not IMP, as IMP recruitment to granules is very low in young neurons.

We thank the referee for this suggestion. We have performed new smFISH experiments to detect endogenous *profilin* RNA in a Me31B-GFP context. These experiments confirmed that the fraction of *profilin* RNAs localizing to granules increases upon aging (see revised Figure 7A). The observed fold change increase, however, is lower (1.5-2 fold increase) than that observed when using Imp as a marker of granules, which is consistent with the fact that it is less affected by the very low number of Imp+ granules found in young brains, as rightly pointed out by the referee.

The fraction of endogenous and reporter mRNA in granules should also be calculated.

To not only assess the recruitment of endogenous RNA to granules, but also the recruitment of reporter RNAs, we quantified the relative proportion of reporter RNAs co-localizing with Me31B in young and aged

brains (Figure 7B). Increased recruitment to granules was observed for both endogenous and reporter RNA upon aging.

As explained in our answer to Referee 3's point 3, quantification of the absolute fraction of RNAs localized to RNP granules can however not rigorously be performed. A rough estimation of the percentage of *profilin* RNA molecules co-localizing with granules is provided in revised Supplementary Figure S7C, though.

-For the quantification of GFP intensities, it may be good to back them using Western blots or other.

In our experiments, *gfp*-3'UTR constructs are expressed specifically in MB neurons, *i.e.* in a very small fraction of the adult fly brain. GFP amounts are thus too low to be visualized by Western Blot performed from entire brain lysates. In our hands, quantifications of GFP signals are much more reproducible when directly measuring intensity of GFP fluorescence than when measuring signals produced *via* anti-GFP immuno-staining and this is why we proceeded that way. At least three replicates were performed for each condition.

Reviewer #3 (Remarks to the Author):

This manuscript focuses on the formation of neuronal granules by the conserved DEAD-box helicase, Me31B in response to aging. Interestingly, Me31B levels increase in aging brains, resulting in fewer but larger (amount of Me31B per granule?) RNP granules and this increase depends on PKA. Me31B granules in aged brains more frequently contain Imp protein than Me31B granules in young brains suggesting that Me31B is limiting for the recruitment of client proteins like Imp. Although necessary, increasing Me31B levels isn't sufficient to form larger granules and promote recruitment of Imp, however, suggesting that there is some other feature of aging that is required that is not addressed here. The authors refer to these Me31B and Imp containing granules as poorly diversified, although this is an overstatement given that they only examine one client protein. Increased association of Imp with Me31B correlates with a decrease in the level of translational reporter for an Imp target RNA, profilin, suggesting that the age-dependent incorporation of Imp into these granules is a mechanism to repress its translation.

My main concern is that the key finding that relates to aging is that Me31B levels increase but nothing in the manuscript addresses how this happens.

The discovery that Me31B levels increase upon aging is indeed an important finding of our manuscript. This finding raises different questions: 1) Is the increased dosage of Me31B related to changes in the properties of Me31B-containing granules upon aging?, 2) is it causing the observed changes? and 3) how is the change in Me31B dosage regulated? In our manuscript, we focused on demonstrating that the observed change in Me31B levels has a biological impact (questions 1 and 2) and did not explore the regulatory aspects. In the revised version of the manuscript, however, we now provide evidence that the age-dependent increase in Me31B levels does not correlate with an increase in *me31B* RNA levels (revised Figure 4C), and thus likely reflects post-transcriptional regulation. This is discussed in the revised version of the discussion.

Instead the authors focus on the idea that granules are being remodeled and changing their functions, but there is little evidence for any remodeling of change in function going on. Furthermore, perhaps this is a semantic issue, but the way that the manuscript is written seems to suggest that Me31B is acting to assemble different types of granules in young versus aged brains. A simpler interpretation is that Me31B assembles one type of granule that recruits varying amounts of Imp from the cytoplasmic pool, depending on the amount of Me31B.

Finally, there is no evidence that the function of the Me31B granules actually changes during the aging process – they may always be repressive for Imp translation, there is just a shift from the amount of Imp in the cytoplasmic translating pool relative to the amount in Me31B granules in the young versus aged brain.

We agree with the Referee's interpretations, regarding granule type, granule function as well as recruitment of Imp protein and realized that the terms/vocabulary used in our initial version have led to misunderstanding. In the revised text, we have thus simplified the terminology used and clarified our interpretations of the observed changes in granule properties.

Specific Issues

1) It isn't clear how the authors are defining granule size (large vs small) – are they referring to the measured size of the granule or the fluorescence intensity (which would relate to number of molecules)? This should be clarified.

Granule size was calculated based on the area (in pixels) of the objects detected using our detection algorithm. It is not linked to fluorescence intensity. To clarify this point, we have implemented the corresponding paragraph of our “Image Analysis” Materials and Methods section.

2) From the images shown, it is not so clear that there are more Imp particles in aged versus young brains but rather there are fewer, brighter spots in aged brains.

It is indeed likely that some low-intensity Imp+ granules are masked in young brains by the high diffuse cytoplasmic signal, and this is the reason why we referred to the observed change as “an increase in the number of Imp-enriched granules **detectable over the cytoplasmic pool**”. To clarify this point further, we have now directly linked this sentence to the finding that the cytoplasmic diffuse signal decreases upon aging.

3) Colocalization of *profilin* RNA and Imp seems very infrequent and there is no quantification. Is there a quantifiable change in granule association of *profilin* in aged versus young brains?

As shown in our initial Figure 7A, a strong increase in the association of *profilin* mRNA with Imp-positive RNP granules is observed upon aging, a result we have now confirmed using Me31B-GFP as a marker (Figure 7A). As noted by the referee, only a fraction of *profilin* RNA molecules seems to associate with granules. The absolute fraction of *profilin* RNA molecules found in granules can however not rigorously be quantified, as only granules above the imaging resolution limit can be detected. Furthermore, detection of granules from images is very sensitive to the threshold used when detecting objects and to the method and parameters used to estimate co-localization. For this reason, we aimed at establishing a robust analysis pipeline with fixed, and not manually-determined, detection parameters. Although this pipeline has a tendency to underestimate co-localization, it is very well adapted to side-by-side comparisons and this is the reason why we reasoned in terms of fold-changes rather than absolute percentage of colocalization in our experiments. We agree however that not stating that only a fraction of *profilin* RNAs localize to Imp-positive granules may have been misleading and we thus have estimated manually, in a subset of images, the fraction of *profilin* RNA molecules found in detectable granules in aged brains of wild-type flies. That the recruitment of *profilin* RNA to granules is not complete is now mentioned in the text and the manually estimated percentage of co-localization shown in revised Supplementary Figure S7C.

4) Based on FRAP experiments, Me31B containing granules seem to be more liquid-like and Imp less liquid-like based on FRAP experiments and the properties don't seem to change between young and aged brains. However, in aged brains, most Me31B also contain Imp. When the authors performed FRAP experiments to analyze Imp in aged brains, were these only performed on Me31B- Imp+ granules or were these performed on Me31B+ Imp+ granules? If the latter (or a mixture), then it would seem that Me31B is not determining granule properties since Imp granules remain less liquid-like in aged brains.

We thank the referee for his/her comment. First, we would like to highlight that the dynamic behavior of individual RNP components is not to be taken as a read-out of the global material properties of entire RNP granules. As already shown for various types of RNP granules, indeed, different protein species belonging to a same condensate can have widely different FRAP kinetics. For example, client proteins are usually more dynamic than scaffolding components. Furthermore, multiple phases with different material properties have already been shown to coexist within the same condensates. Second, Me31B- Imp+ granules were never observed in our experiments. Only two types were observed: Me31B+ Imp+ and Me31B+ Imp- granules. Thus, the FRAP experiments performed in aged flies were all performed on granules containing both Me31B and Imp.

5) I am not convinced by the claim that there is a change in the sorting of RNP components to distinct granules or that Me31B and Imp condense into large multiphase granules in aged brains.

Imp does associate with Me31B granules in young brains and the increase in the Me31B+ Imp+ granules from young to old is only several fold.

Here, we feel that the referee's concern is largely of semantic origin. As illustrated in Figure 3A, both Me31B+ Imp⁻ and Me31B+ Imp⁺ granules can be detected in young, but only Me31B+ Imp⁺ are detected in aged brains. The presence of these two categories of granules in young flies led us to conclude about a differential sorting of Imp molecules, a process that, as suggested by the referee, reflects the efficient recruitment of Imp to some, but not all granules. To clarify this point, we have modified our initial description of the differential recruitment of Imp to Me31B+ granules (see revised manuscript, pages 7-8).

Why a 2-to-3 fold increase in the proportion of Me31B+ Imp⁺ granules is not convincing to the referee is however unclear to us, and we hope that the new graph presented in revised Figure 3C better represents the proportion of the different classes of granules and their evolution upon aging.

Why the referee is "not convinced that Me31B and Imp condense into large multiphase granules in aged brains" is also unclear to us, as we have shown that i- granules are larger and brighter in aged flies than in young ones, (Figure 1) ii- granules found in aged flies all contain both Imp and Me31B (unlike in young flies) (Figure 3B,C), and iii- Imp and Me31B localize to distinct sub-compartments (Figure 3D and movie 1).

Do the Me31B granules in young brains that contain Imp have more Me31B than the granules that don't contain Imp? It seems equally possible that as the amount of Me31B increases with aging, more Imp is recruited from the cytoplasmic pool into Me31B granules, which then just increases the frequency of finding Imp in Me31B granules.

We thank the Referee for his/her suggestion. To test whether the amount of Imp recruited to a given Me31B+ granule depends on the amount of Me31B present in that granule, we plotted the average intensity of Me31B signal in Me31B+ Imp⁺ and Me31B+ Imp⁻ granules of young brains. As shown in our revised Figure 3E, a significantly higher intensity is observed in Me31B+ Imp⁺ granules, indeed suggesting that the amount of Me31B might be a key factor underlying the differential recruitment of Imp. We don't think that these results are in contradiction to the idea of differential sorting, but rather that they provide a mechanistic explanation underlying this process.

Related to this and the above point, how does the frequency of Me31B- Imp⁺ granules change? Do all Imp⁺ granules contain Me31B in aged brains?

As explained in our answer to point 4, only two types of granules (Me31B+ Imp⁺ and Me31B+ Imp⁻) could be observed in our assays. Me31B- Imp⁺ granules were neither observed in young nor in aged brains. We hope the new graph shown in Figure 3C will clarify this point.

6) Was the experiment in Figure 4 performed using young or aged brains? Is Me31B required for granules to form in young brains or just for the larger granules in aged brains? This isn't shown and should be made clear.

We thank the referee for pointing this out. The results shown in the initial Figure 4 were obtained in aged brains, and this information is now provided in the corresponding legend of the Figure, which is now found in revised Supplementary Figure S3.

In addition, it isn't clear why the authors conclude that inactivation of Me31B prevents assembly of Imp-containing granules – rather it seems to prevent the formation of Me31B granules that recruit (see point #5).

Indeed, assembly of Me31B+ granules is lost in this context and we have modified our description in the revised text.

7) In Figure 5, when Me31B is reduced, it seems like colocalization of Imp with Me31B is not affected – does the level of Me31B still show an increase in aged brains?

We have measured the amount of Me31B proteins in young and aged *me31B Δ 1/+* and *me31B Δ 2/+* flies. As expected, a 50% decrease in the total amount of Me31B was observed when comparing control and *me31B Δ 1/+* flies (revised Supplementary Figure S4A). Furthermore, a 1.5 fold increase in Me31B levels was observed upon aging in these flies (revised Supplementary Figure S4B), similarly to the control condition.

Furthermore, it is difficult to compare the data in Figures 5D and 5E, which is necessary to be able to interpret the effect of reducing the level of Me31B on Imp. A cumulative frequency distribution for Imp should be shown and, vice versa, the number of granules plotted for Me31B similarly to E.

As the number of individual granules is very high, we had chosen to plot the distribution of granule size both as cumulative frequency distributions and as scatter dot plots. As requested by the reviewer, we have also generated the two types of graphs for Imp granule numbers. In the revised version of the manuscript, we provide for both parameters the frequency distributions in the main Figures (revised Figure 5D and E) and the scatter dot plots in the Supplementary Figures (revised Supplementary Figures S5A and D).

Also related to the experiment in Figure 5, the conclusion that "MB gamma neurons tune the concentration of limiting scaffold protein Me31B on aging so as to trigger its condensation into large granules. This only mildly promoted recruitment of Imp to Me31B granules...." is contradictory to data presented earlier showing that nearly all Me31B granules contain Imp in aged brains.

We don't think this is contradictory with the fact that all Me31B granules contain Imp in aged brains. Rather, it indicates that increasing the dosage of Me31B is not sufficient to trigger recruitment of Imp. Consistent with this idea, our results indicate that PKA activity is also required.

8) Rather than PKA preventing condensation of Imp into larger granules it seems equally plausible that PKA prevents the incorporation of Imp into Me31B granules.

We are not sure about the conceptual differences implied by the Referee for the two proposed hypotheses, but agree with the idea that PKA promotes (rather than prevents) the incorporation of Imp into Me31B+ granules, a conclusion we had made at the end of the section describing PKA function.

9) Figure 7C is not referred to in the text. Perhaps the image is supposed to illustrate colocalization of profilin with Imp quantified in Figure 7A but there doesn't seem to be much evidence for colocalization based on the image.

We thank the Referee for pointing this out. In the revised version of the manuscript, this panel has moved to revised Supplementary Figure S7A and is now referred to in the main text. Furthermore, we clarified in the main text that only a fraction of *profilin* RNAs co-localizes with granules and show in revised Supplementary Figure S7C a manual estimation of the percentage of co-localization. Noteworthy, as explained in our answer to point 3, this percentage is under-estimated, as granules below the detection limit are missed.

10) Is the *gfp-profilin* 3'UTR also localized to granules – this should be shown, especially because the image in Figure 7C is not convincing.

We performed new smFISH experiments to visualize co-localization between *gfp-profilin* 3'UTR RNAs and RNP granules (Figure 7B). As seen for endogenous *profilin* mRNA, only a fraction of *gfp-profilin* 3'UTR RNAs co-localized with Me31B. This fraction, however, is still significantly higher than that expected from random co-localization. Furthermore, it significantly increases upon aging (Figure 7B).

11) Why rely on a reporter for profilin translation rather than detecting Profilin levels directly?

Detecting Profilin levels directly is a complementary way to analyze *profilin* protein product and this is the reason why we had analyzed Profilin levels upon aging and provided the corresponding information in our initial Supplementary Figure S5B (now revised Supplementary Figure S8A). Although measuring Profilin levels has the advantage of providing a readout of endogenous protein levels, it is not a proper readout of translation *per se*, as Profilin protein stability/turnover may be modified upon aging. Comparing the levels of a single protein (GFP), expressed from different reporters prevents such caveats.

Additionally, how can the authors distinguish whether the change in translation is due to granule association versus just an effect of age? Performing the experiment in an *me31B* knockdown where the granules don't form would be more informative.

We thank the referee for this suggestion. To strengthen the hypothesis that condensation plays an important role in translational repression, we inactivated *me31B* via RNAi and analyzed the translation of *GFP-profilin* 3'UTR upon aging in this context. As shown in revised Figure 8A, no significant decrease in GFP expression could be observed in this context, thus establishing a causal link between condensation and translation repression.

Although the PKA experiment is an attempt to do this, PKA could have other effects on translational activity independent of granules.

This is indeed an important point that we addressed by monitoring the effect of PKA on the translation of non granule-associated mRNA species (SV40 and *camk2* 3'UTR reporters). As shown in our revised Figure 8C and Supplementary Figure S9B, inhibition of PKA activity did not increase the translation of SV40 and *camk2* RNAs, indicating that translation is not generally increased in this context, and that the translational repression of *profilin* mediated by PKA is specific.

Minor points

1) The word "cluster" seems like an odd choice to refer to the accumulation of RNP components in granules.

We have modified the text according to the Referee's suggestion, avoiding the use of the term "cluster".

2) How do the authors know that the granules they observed in aged brains aren't pathological?

We agree with the Referee that we have not specifically tested the pathogenicity of the large granules found in aged brains and have thus removed our statement about their non-pathogenicity.

3) Why is it important to repress *profilin* translation in aged brains? The authors should comment on this in the Discussion.

This is an interesting question that we have now in part addressed by analyzing the organization of the F-actin cytoskeleton upon aging (see answer to point 3 of Referee 1). This aspect is now mentioned in our revised discussion.

Reviewer #4 (Remarks to the Author):

The manuscript by the Besse group examines the dynamics of RNP granules in the physiological context of the aging *Drosophila* brain. The study first compares known granule markers (Imp and Me31B, among others) between young and aged *Drosophila* brains and describes a specific enrichment of these markers in granules upon aging. Next, the authors provide evidence that aging-associated granules are dynamic and result from the coalescence of smaller pre-existing granules. The authors then link these granules to an increase in Me31B dosage observed during normal aging and suggest they are sites of age-dependent localized translational repression. This repression is sensitive to PKA inactivation, which also differentially reduces Imp (strongly) and Me31 (mildly) recruitment to granules, providing a link between RNP component recruitment and translational repression in condensates.

This manuscript documents for the first time, to my knowledge, the condensation of RBPs in granules in the context of normal aging. I find this interesting and promising in which it adds novel physiological depths to the field. However, the manuscript falls short in exploring the functional consequences of this condensation. In particular, the GFP-3'UTR reporter is a poor choice to monitor localized translation (especially since alternative, more recent, systems exist: for example, the SunTag system recently used by the authors PMID: 33890854). Moreover, experiments based on PKA inactivation provide weak functional evidence due to the many non-specific effects of such inactivation.

Major comments

Condensation characterisation:

1. In Fig. S1, the authors claim that certain RBPs (Tral, HPat1, Stau) cluster upon aging while others do not (PABP and Rin). This conclusion needs to be supported with quantitative measurements.

In the revised version of Supplementary Figure S1, we now provide plots showing the distribution of Tral+,

HPat1+ and Stau+ granule numbers in both young and aged neurons. Significant changes in granule number were observed for all markers tested (Figure S1K-L).

2. The authors set out to investigate if the Me31B+ Imp+ RNP granules observed upon aging result from the merging of two types (Me31B+ Imp+ or Me31B+ Imp-) of pre-existing granules. To address this, they quantify the fraction of Me31B+ Imp+ granules to Me31B+ Imp- in aged and young flies and deduce that an increase in ratio must result from changes in the trafficking of granule RNP components upon aging. However, this observation can simply be the result from new diffuse Imp localizing in granules rather than two distinct types of granules coalescing. Live imaging would provide a stronger argument for coalescence. However, there is no such coalescence event visible in video 1.

We agree with the Referee that fusion between two types of pre-existing granules may not account for the remodeling of RNP observed upon aging, and this is precisely why we used the term coalescence exclusively when referring to components, not granules (“coalescence of components initially sorted to distinct granules”). As such vocabulary issues have been misleading for Referee 3 also, we have thus modified our description of the differential recruitment of Imp to Me31B+ granules (see revised manuscript, page 7), clarifying that we do not imply coalescence of initially distinct granules.

Regarding live-imaging, we have indeed been able to visualize fusions between Me31B+ granules using single-color imaging (Formicola et al. 2021). As pointed out by the Referee, however, these events are rare and reproducibly monitoring fusion events between Me31B+ Imp- and Me31B+ Imp+ would require imaging with high temporal resolution for extensive periods of time, which is currently not compatible with preserving the integrity of brain explants.

3. In Fig. 3, can the authors please also provide the proportion of Me31B+ Imp+ and Me31B+ Imp- granules in young and aged flies rather than just the aged:young ratio? This will improve data clarity and readability.

We have modified the graph so as to better represent age-dependent changes in the proportion of Me31B+ Imp+ and Me31B+ Imp- granules. We also have added the Me31B- Imp+ category in the graph legend so as to better highlight that this latter class of granules is not observed in our samples.

Me31B dosage study:

4. The authors claim that RNAi-mediated knockdown of me31B but not Imp prevents the assembly of granules. Please provide the same granule number quantification for the Imp RNAi experiment to support this claim.

The corresponding graph is provided below (Figure D). As we extensively modified the text and decided to remove the idea of scaffold and client, this result is however not mentioned in the revised version of the manuscript.

Figure D. Inactivating *imp* does not modify the number of Me31B+ granules. Two replicates were performed and the mean value of each replicate is indicated as a symbol (triangle). Data points were color-coded based on the experimental replicate they belong to. ns stands for not significant. n.s. stands for not significant (unpaired t-test).

5. To prove that Me31B dosage affects its condensation, the authors remove a copy of me31B and observe that Me31B granules in aged flies resemble the young phenotype. The reduction of Me31B granule size can however be due to the lower levels of Me31B expression rather than a change in compartmentalization (i.e. less cytoplasmic Me31B available overall, and not a change in Me31B's localization). Comparing the partition coefficient of Me31B between WT and the two deletion lines would

be informative in this case.

We thank the referee for this suggestion, which made us re-evaluate Me31B partition coefficients using the Me31B-GFP knock-in line we generated. The new quantifications we performed using this line first revealed that Me31B partition coefficient does actually not increase upon aging, as we initially described based on experiments performed using antibody staining. Given that i- signals resulting from antibody stainings are much more variable than those obtained by directly measuring endogenous fluorescence intensity, and that ii- only 2 of the 3 replicates performed using immuno-staining were indicative of an increased partition coefficient, while all new replicates (6/6) performed based on Me31B-GFP signal did not point to any increase in partition coefficient, we decided to revise our statement and conclusions. In the revised Figure 4A, we thus now show that Me31B partition coefficient does not increase upon aging. As suggested by the Referee, this indeed indicates that changes in Me31B levels, rather than changes in partition coefficient, explain Me31B condensation upon aging.

To address the Referee's question, we plotted Me31B-GFP partition coefficient in *me31BΔ1/+* and *me31BΔ2/+* backgrounds (Figure E). As expected from our new results, no difference in the partition coefficient of Me31B was observed in these contexts when compared to controls.

Figure E. Partition coefficients of Me31B-GFP in control and *me31BΔ* contexts.

No significant difference could be observed when removing a copy of *me31B*.

Noteworthy, the referee's comment, together with these new results, led us to extensively re-organize our results (pages 7-9) and Figures (Figures 1, 4 and 5). We hope this helped clarify our description of age-dependent phenotypes.

6. Please provide quantitative evidence in this manuscript that removing a copy of the Me31B gene reduced stable levels of the protein.

As suggested by the referee, we have analyzed the levels of Me31B in *me31BΔ/+* conditions and confirmed that removing one copy of *me31B* indeed induces a 50% decrease in the levels of Me31B protein (see revised Supplementary Figure S4A).

GFP translation reporter

7. The authors use a GFP reporter fused to the 3'UTR of the granule enriched RNA profilin or two other control 3'UTRs (those of SV40 and the non-granule enriched transcript *camk2*) as a tool to measure the translational regulation in granules. This approach to study localized translation is questionable and has several issues:

(i) the GFP- profilin 3' UTR construct was not shown to localize in granules ,

We thank the reviewer for his/her suggestion. We have performed and analyzed co-localization experiments and showed that *gfp-profilin* 3'UTR partially, but significantly, co-localizes with granule markers (revised Figure 7B).

(ii) total GFP fluorescence is not an indicator of translational output per se, but rather protein steady state levels that can be linked to other processes such as protein stability or turnover (even if RNA levels revealed by smFISH remain similar),

We agree with the referee that measuring GFP fluorescence does not reflect the dynamics of translation, but rather a cumulated number of translation products. We however believe that the low temporal

resolution provided by these reporters is still compatible with a comparison of two time points separated by more than 30 days (young and aged flies).

We also agree with the referee that the stability or turnover of GFP molecules impacts on GFP signal independently of translation efficiency. The characteristic stability and turnover of GFP molecules produced will however be identical for all reporters and thus cannot explain the differences observed between reporters.

(iii) even if translation inhibition was taking place upon aging, it may not be exclusive to granule RNAs, as suppression of non-granule localized GFP transcripts in the cytosol will also contribute to the total reduction in fluorescence.

The referee is right and we realized that our reference to “granule-associated mRNAs” was incorrect, as we were referring to mRNAs species that show localization to granules, not necessarily to RNA molecules that were actually associated with granules. We have corrected this in the text.

8. To directly demonstrate localized translational repression in granules, the authors should take advantage of the SunTag system recently used by them in the same organism.

We agree with the referee that this would indeed be very nice. This, however, is beyond current technical limitations, as the *SunTag-profilin* constructs we previously used (Formicola et al. 2021) allowed us to detect translational bursts induced by neuronal activation, but does not support detection of endogenous translation events in resting neurons. Despite many attempts, and for reasons we don't completely understand, no ScFv-GFP+ foci exhibiting i-co-localization with *profilin* smFISH signals, and ii- sensitivity to Puromycin treatment could be detected in either young or aged resting neurons.

We however think that our new experiment, where we prevent Imp condensation through RNAi-inactivation of *me31B* (revised Supplementary Figure S3A-D), points to a role of condensation in translational repression. In this experiment, indeed, no age-dependent repression of the *gfp-profilin* 3'UTR reporters could be observed (Figure 8A).

Role of PKA

9. The authors notice a translational de-repression upon inactivating PKA. Since PKA repression was associated with a reduction of RBP recruitment in granules (significant for Imp but mild for Me31B), the authors suggest that granule-localized Imp must be critical for translational regulation there. In addition to having the same issues of using GFP steady state fluorescence to monitor translation, the approach of inhibiting PKA is not specific enough and can alter many cellular pathways. Thus, the observed effects on GFP fluorescence levels cannot be solely attributed to lower levels of Imp in granules. For instance, Me31B is also mildly affected by the PKA inactivation.

Three experiments were performed to address the specificity of PKA down-regulation on RNP components and translational regulation:

1) Treatment of aged brain explants with the H-89 inhibitory compound, to acutely inhibit PKA and thus prevent long-term indirect effects. This revealed that H-89 triggers the re-localization of Imp from the granules to the cytoplasm within dozens of minutes (revised Supplementary Figure S6C,D).

2) Analysis of the effect of PKA inactivation on additional granule markers. This revealed that PKA inhibition does not prevent the recruitment of Tral and Hpat1 to granules, further strengthening the specificity of the effect on Imp (revised Supplementary Figure S6A,B).

3) Analysis of the effect of PKA inhibition on non-granule associated RNA species. This revealed that inhibition of PKA does not increase the translation of non granule-associated mRNAs (Figure 8C and Supplementary Figure S9B), and thus that PKA specifically promotes the repression of *profilin* translation.

Minor comments

1. In the third paragraph of the introduction, the authors state that RNP condensates have up to dozens of RNA and protein molecules. However, recent transcriptomic and proteomic approaches (that the authors cite) suggest a much higher number. Please clarify this sentence.

To our knowledge, the stoichiometry of individual RNP granules still remains to be determined as transcriptomic and proteomic approaches did not address the question of the number of molecules per

granule. We however understand the Referee's point and have replaced the term "up to dozens" by "up to hundreds".

2. It would be good if the authors can better explain how partition coefficients are calculated in the material and methods section since they are essential to several conclusions of the manuscript.

We have implemented our description of how partition coefficients were calculated in the revised version of the materials and methods section.

3. In Fig. 1L, Does the young condition have only one replicate?

No, we had performed three independent experiments, each with young and aged replicates. As representing different replicates on a cumulative frequency distribution is difficult, we added in the revised Supplementary Figure S1A distribution plots displaying the mean of individual replicates (triangles).

Is this the cumulative percentage curve?

Yes, values on the y axis represents percentages and we have now added this information on the graph axis.

4. Live imaging description in the methods section states that the brains of 5-day old flies are imaged, whereas the legend of video 1 states that it was 10-day old flies.

We thank the referee for pointing this out. Flies were 5 day-old and the legend to Movie 1 has been corrected.

References

- Formicola N, Heim M, Dufourt J, Lancelot AS, Nakamura A, Lagha M, Besse F. 2021. Tyramine induces dynamic RNP granule remodeling and translation activation in the *Drosophila* brain. *eLife* **10**.
- Hillebrand J, Pan K, Kokaram A, Barbee S, Parker R, Ramaswami M. 2010. The Me31B DEAD-Box Helicase Localizes to Postsynaptic Foci and Regulates Expression of a CaMKII Reporter mRNA in Dendrites of *Drosophila* Olfactory Projection Neurons. *Front Neural Circuits* **4**: 121.
- Sears JC, Broadie K. 2020. FMRP-PKA Activity Negative Feedback Regulates RNA Binding-Dependent Fibrillation in Brain Learning and Memory Circuitry. *Cell Rep* **33**: 108266.
- Zhang Q, Huang H, Zhang L, Wu R, Chung CI, Zhang SQ, Torra J, Schepis A, Coughlin SR, Kornberg TB et al. 2018. Visualizing Dynamics of Cell Signaling In Vivo with a Phase Separation-Based Kinase Reporter. *Molecular cell* **69**: 347.

REVIEWERS' COMMENTS

Reviewer #1 (Remarks to the Author):

This is an extensively revised version from the previous manuscript. All my comments/concerns have been addressed or answered.

Reviewer #2 (Remarks to the Author):

In the extensive revision of their manuscript, the authors have performed a number of experiments to address the reviewer's comments.

The authors still don't have mechanistic explanation for the action of PKA or the events that lead to increased Me31B expression. The authors however nicely show that translational repression is a consequence of granule formation and not the opposite, although the effects are mild. Overall, the authors show in a physiological system a functional role of Me31B condensates and how this regulates gene expression during aging, and I support publication of the manuscript.

Comment:

The authors should include in the main text the fraction of profilin mRNA that localizes in Me31B granules (Figure S7C; ~15%) and discuss this as it is an important piece of data with mechanistic/functional consequences. Note that the value they found is in the range of what has been found in many other systems and in particular mammals.

Reviewer #3 (Remarks to the Author):

The authors have satisfactorily addressed my concerns in the revised manuscript.

Reviewer #4 (Remarks to the Author):

Pushpalatha et al. did extensive effort in revising this work. The new observation that the partition coefficient of Me31B does not change upon aging is critical and considerably changes the view of Me31B condensation which caused the authors to correct several unwarranted conclusions. Moreover, experiments involving PKA inhibition and monitoring the translation of a reporter are solidified with the addition of crucial controls.

This is now a stronger manuscript: experiments are better designed, and conclusions are reasonably supported by the data. I have a few comments, which can probably be addressed by textual changes, before I can recommend the paper for publication.

Major comments:

1. It is now clear that aging accompanies an increase in Me31B protein expression but without a change in the proportion of protein localized in granules. This results in more Me31B in granules overall which, not surprisingly, increases their size. The effect of this on granule recruitment of Imp is however nuanced. On one hand, Me31B promotes Imp recruitment as evidenced by the observations that Me31B+ Imp+ granules have more Me31B than Me31B+ Imp- ones, and RNAi against Me31B+ reduces Imp recruitment to granules. On the other hand, me31B deletion mutants (with half the amount of protein) do not have altered Imp granule recruitment. How can the authors consolidate these two notions? It is likely that RNAi lowers protein levels much more than the mutants. If so, this should be clearly stated and emphasized in the main text.

2. In Figure S4b, do the WT and mutants with a deleted copy of me31B have a similar amount of me31B protein in the aged flies?

Minor comments:

1. In Figure 1n (and elsewhere where applicable), please change the title of the Y-axis since the graph represents measured averaged fluorescent intensities (a readout of concentration) rather than the concentration itself.

2. Similarly, the authors can invest effort in making their graph axes more readable. For example, "Protein amount" on the Y-axis of Figure 4b is not very informative.

3. In line3 of page 8, the authors state that ALL Me31B+ granules contain Imp in the aged brain. However, I can see some Me31B+ granules in panel 3b'' without Imp. Perhaps it would be more accurate and cautious to state that Me31B+ granules predominantly contain Imp.

4. I suggest being specific in the paragraph title in page 7: Me31B granule compositional diversity decreases upon aging.

5. In line7 of page 8, the phrase "This process is accompanied by the loss of Me31B+ Imp- granules" gives the impression that such granules dissociated. Me31B+ Imp- granule can persist, but with Imp joining upon aging. I suggest rewriting or moving the phrase in question.

6. In page 8, the paragraph that aims to "investigate the origin of the age-dependent changes in Me31B behaviour" needs to have a concluding sentence.

7. In page 9, I recommend changing "Me31B is specifically regulated upon physiological aging" into "Me31B is more expressed upon physiological aging" which is more explicit.

8. It would be good to include Figure E of the authors' response in the main text if possible. This will further show that the partition coefficient is stable even in the mutant background.

9. In Figure S7b, the graph's y-axis reads "# RNA in granules" while its legend says "Relative fractions of RNA molecules found in granules. In addition, if this number is calculated per imaging field, please do state so on the graph's axis.

Reviewer #1:

This is an extensively revised version from the previous manuscript. All my comments/concerns have been addressed or answered.

Reviewer #2:

In the extensive revision of their manuscript, the authors have performed a number of experiments to address the reviewer's comments.

The authors still don't have mechanistic explanation for the action of PKA or the events that lead to increased Me31B expression. The authors however nicely show that translational repression is a consequence of granule formation and not the opposite, although the effects are mild. Overall, the authors show in a physiological system a functional role of Me31B condensates and how this regulates gene expression during aging, and I support publication of the manuscript.

Comment:

The authors should include in the main text the fraction of profilin mRNA that localizes in Me31B granules (Figure S7C; ~15%) and discuss this as it is an important piece of data with mechanistic/functional consequences. Note that the value they found is in the range of what has been found in many other systems and in particular mammals.

We thank the referee for this suggestion and are now explicitly mentioning the fraction value in the text of our revised discussion (page 15). We also compare this value to what has been seen in mammalian systems and discuss about the implications about this finding.

Reviewer #3:

The authors have satisfactorily addressed my concerns in the revised manuscript.

Reviewer #4:

Pushpalatha et al. did extensive effort in revising this work. The new observation that the partition coefficient of Me31B does not change upon aging is critical and considerably changes the view of Me31B condensation which caused the authors to correct several unwarranted conclusions. Moreover, experiments involving PKA inhibition and monitoring the translation of a reporter are solidified with the addition of crucial controls.

This is now a stronger manuscript: experiments are better designed, and conclusions are reasonably supported by the data. I have a few comments, which can probably be addressed by textual changes, before I can recommend the paper for publication.

Major comments:

1. It is now clear that aging accompanies an increase in Me31B protein expression but without a change in the proportion of protein localized in granules. This results in more Me31B in granules overall which, not surprisingly, increases their size. The effect of this on granule recruitment of Imp is however nuanced. On one hand, Me31B promotes Imp recruitment as evidenced by the observations that Me31B+ Imp+ granules have more Me31B than Me31B+ Imp- ones, and RNAi against Me31B+ reduces Imp recruitment to granules. On the other hand, me31B deletion mutants (with half the amount of protein) do not have altered Imp granule recruitment. How can the authors consolidate these two notions? It is likely that RNAi lowers protein levels much more than the mutants. If so, this should be clearly stated and emphasized in the main text.

We thank the referee for pointing this out. Indeed, RNAi against *me31B* lowers Me31B protein levels much more than removing a copy of the *me31B* locus. Thus, we think that there is enough Me31B in granules in the *me31BΔ/+* context to recruit Imp, but not upon *me31B* RNAi. As the extent of *me31B* inactivation upon RNAi could only be appreciated by looking at the images shown in Supplementary Fig. 3A and C, we more precisely quantified changes in Me31B levels. This revealed a four-fold decrease in Me31B levels upon RNAi (revised Supplementary Fig. 3a), compared to the two-fold decrease observed when removing one copy of *me31B*.

2. In Figure S4b, do the WT and mutants with a deleted copy of me31B have a similar amount of me31B protein in the aged flies?

No, WT individuals have approximately twice more Me31B than *me31BΔ/+* ones (Supplementary

Figure S4A).

Minor comments:

1. In Figure 1n (and elsewhere where applicable), please change the title of the Y-axis since the graph represents measured averaged fluorescent intensities (a readout of concentration) rather than the concentration itself.

Axis legends have been changed in Figure 1n, Figure 3e, Figure 5a

2. Similarly, the authors can invest effort in making their graph axes more readable. For example, "Protein amount" on the Y-axis of Figure 4b is not very informative.

We have changed the axes of Figure 4b, Supplementary Figure S3e, Supplementary Figure S4a,b, and Supplementary Figure S5c.

3. In line3 of page 8, the authors state that ALL Me31B+ granules contain Imp in the aged brain. However, I can see some Me31B+ granules in panel 3b" without Imp. Perhaps it would be more accurate and cautious to state that Me31B+ granules predominantly contain Imp.

We thank the referee for this comment, which made us realize that there might indeed be some confusion when looking at Figure 3b. Some Me31B+ Imp- granules are indeed visible in the lower left corner of the image, but they belong to non-MB γ neurons that do not express Imp. We have now added an asterisk pointing to those cells and a note in the corresponding legend. In addition, as we cannot exclude that a very small proportion of Me31B+ granules are Imp- in old brains, we have modified our text into "almost all Me31B+ granules ...".

4. I suggest being specific in the paragraph title in page 7: Me31B granule compositional diversity decreases upon aging.

We have changed the title into "Me31B and Imp condense into common multiphase condensates upon aging."

5. In line7 of page 8, the phrase "This process is accompanied by the loss of Me31B+ Imp- granules" gives the impression that such granules dissociated. Me31B+ Imp- granule can persist, but with Imp joining upon aging. I suggest rewriting or moving the phrase in question.

We have rewritten this part and deleted the problematic sentence.

6. In page 8, the paragraph that aims to "investigate the origin of the age-dependent changes in Me31B behaviour" needs to have a concluding sentence.

We have modified the last sentence of this paragraph to provide a clearer conclusion.

7. In page 9, I recommend changing "Me31B is specifically regulated upon physiological aging" into "Me31B is more expressed upon physiological aging" which is more explicit.

We have replaced the terms "is specifically regulated" by "increases".

8. It would be good to include Figure E of the authors' response in the main text if possible. This will further show that the partition coefficient is stable even in the mutant background.

We have inserted the Figure E of the authors' response in Supplementary Fig. 4c and are referring to it in the main text (page 9).

9. In Figure S7b, the graph's y-axis reads "# RNA in granules" while its legend says "Relative fractions of RNA molecules found in granules. In addition, if this number is calculated per imaging field, please do state so on the graph's axis.

We thank the referee for raising this point. We have now relabeled the y axis of the graph in Figure S7b as "Fraction of RNA in granules". Furthermore, we now specify in the Figure legend that the percentage of co-localization was calculated for each imaged field.